# Airway basal cells show a dedifferentiated KRT17[high]Phenotype and promote fibrosis in idiopathic pulmonary fibrosis

Benedikt Jaeger[1,2,15], Jonas Christian Schupp [2,3,4,15], Linda Plappert [1,2], Oliver Terwolbeck [1,2], Nataliia Artysh [1,2,4], Gian Kayser [5], Peggy Engelhard[6], Taylor Sterling Adams [3], Robert Zweigerdt [7], Henning Kempf [7], Stefan Lienenklaus [8], Wiebke Garrels [8], Irina Nazarenko[9,10], Danny Jonigk [2,11], Malgorzata Wygrecka [12], Denise Klatt [13], Axel Schambach [13,14], Naftali Kaminski [3] & Antje Prasse [1,2,4] ✉

Idiopathic pulmonary fibrosis (IPF) is a fatal disease with limited treatment options. In this study, we focus on the properties of airway basal cells (ABC) obtained from patients with IPF (IPF-ABC). Single cell RNA sequencing (scRNAseq) of bronchial brushes revealed extensive reprogramming of IPF-ABC towards a KRT17[high] PTEN[low] dedifferentiated cell type. In the 3D organoid model, compared to ABC obtained from healthy volunteers, IPF-ABC give rise to more bronchospheres, de novo bronchial structures resembling lung developmental processes, induce fibroblast proliferation and extracellular matrix deposition in co-culture. Intratracheal application of IPF-ABC into minimally injured lungs of Rag2[−/−] or NRG mice causes severe fibrosis, remodeling of the alveolar compartment, and formation of honeycomb cyst-like structures. Connectivity MAP analysis of scRNAseq of bronchial brushings suggested that gene expression changes in IPF-ABC can be reversed by SRC inhibition. After demonstrating enhanced SRC expression and activity in these cells, and in IPF lungs, we tested the effects of saracatinib, a potent SRC inhibitor previously studied in humans. We demonstrate that saracatinib modified in-vitro and in-vivo the profibrotic changes observed in our 3D culture system and novel mouse xenograft model.

Idiopathic pulmonary fibrosis (IPF) is a progressive disease with a lethal prognosis despite the introduction of new anti-fibrotic treatments[1]. The histological pattern of IPF, usual interstitial pneumonia (UIP), is characterized by patchy and peripheral remodeling of the alveolar compartment[2,3] with replacement of the normal alveolar architecture by fibroblastic foci, honeycomb cysts, and distorted airways. Recent data indicate that IPF exhibits features of small airway disease[4]. Bronchiolization, the replacement of the resident alveolar

epithelial cells by ABCs within remodeled regions in the IPF lung[5–8], leads to a dramatic shift in the epithelial cell repertoire of the alveolar compartment. The ABC is the progenitor cell of the airway epithelium and can give rise to any type of airway epithelial cell such as secretory, goblet or ciliated cells[9]. ABCs play an important role in lung development, start the branching and tubing morphogenesis by building up the airway tree[10], and have been implicated in the pathogenesis of chronic obstructive pulmonary disease and lung cancer[11,12]. We have

recently confirmed the abundance of ABCs in the lung parenchyma of patients with IPF and discovered that the presence of an ABC signature in the transcriptome of bronchoalveolar lavage (BAL) cell pellet of patients was indicative of enhanced disease progression and mortality[13]. Recent scRNAseq data of IPF tissues confirmed increase in ABCs of IPF lung tissues and described a unique aberrant basaloid KRT17+ cell population, which lacks the characteristic basal cell marker KRT5[7,14,15]. However, none of these studies tested functional properties of human IPF-ABC. Up to date, it has not been clear whether ABCs contribute to the fibrotic process or simply represent a bystander phenomenon.

In this study, we sought to determine the transcriptional changes of IPF-ABCs and their potential profibrotic properties using several translational models. Single-cell RNA sequencing (scRNAseq) of bronchial brushings demonstrated extensive reprogramming of IPF-ABCs toward a dedifferentiated KRT17^high PTEN^low cell type with high expression of various transcription factors associated with stemness. Using the organoid model, we determined that IPF-ABCs obtained from individuals with IPF (IPF-ABCs) are characterized by significantly increased formation of bronchospheres, de novo bronchial structures resembling lung developmental processes, and by having profibrotic properties on lung fibroblasts. Using a novel in vivo xenograft model, we discovered that ABCs from individuals with IPF have the capacity to augment fibrosis, remodeling, and bronchialization in the mouse lung. Connectivity map analyses revealed enrichment of src signaling in IPF-ABCs. Saracatinib, a known inhibitor of the src kinase pathway, blocked bronchosphere generation in vitro and attenuated bronchialization and fibrosis in vivo.

## Results

### Single-cell sequencing of bronchial brushes identifies reprogramming of IPF-ABCs in contrast to ABCs derived from disease control (NU-ABC)

scRNASeq was performed on cells obtained by bronchial brushing from nine IPF patients and six nonUIP ILD controls (Supplementary Table 1). 14,873 epithelial single-cell transcriptomes were profiled. Based on expression profiles of known marker genes, we identified four epithelial cell populations in the brushed cells (Fig. 1a–c and Supplementary Fig. 1): ciliated cells (FOXJ1, HYDIN, 41.5% of all epithelial cells), secretory cells (SCGB1A1, SCGB3A1, 10.5%), ABCs (TP63, KRT5, 47.1% of cells), and ionocytes (STAP1, PDE1C, 0.8% of cells), and focused our analysis on ABCs.

Gene expression of IPF-ABC was, however, substantially different from NU-ABC (1099 genes at FDR < 0.05, Fig. 1d). The IPF ABCs were characterized by an increased expression of stem cell markers and stemness increasing signal transduction factors such as FOSL1, KLF4, MYC, CD24, SOX4, which was accompanied by a loss of PTEN (Fig. 1d, e and Supplementary Fig. 2). Expression of KRT17, KRT6A, stratifin (SFN), CTGF and genes for integrin subunits such as ITGB6, ITGAV, ITGB1 and ITGB8 was highly increased (Fig. 1d, e and Supplementary Fig. 2). Moreover, an increased expression of the EGF family members AREG and HBEGF and the shedding enzymes for amphiregulin, ADAM17, and for EGFR, ADAM9, was observed (Fig. 1d, e and Supplementary Fig. 2). Pathway analysis showed multiple pathways upregulated related to cancer, mechanotransduction, ECM sensing, cellular senescence, EMT, TNF-α, and IL-17 signaling (Fig. 1f). Of note, while we observed an overlap with the recently described aberrant basaloid

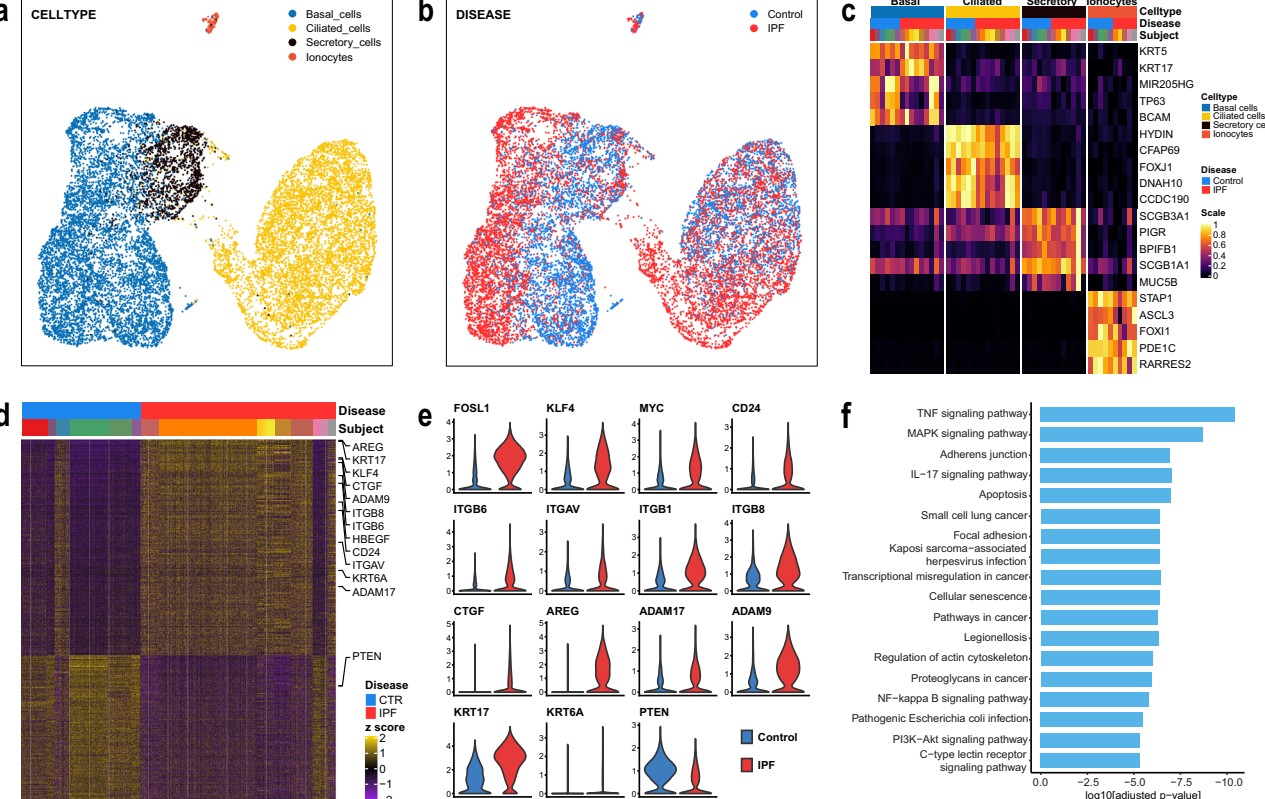

**Fig. 1 | Single-cell RNAseq of bronchial epithelial cells.** scRNAseq was performed on brush cells of IPF patients (*n* = 9) and nonUIP ILD controls (*n* = 6). **a, b** Uniform Manifold Approximation and Projection (UMAP) of *n* = 14,873 single-cell transcriptomes visualizes the four major discrete epithelial cell types (detected UMAPs colored by **a** cell types, **b** disease state). **c** Heatmap of unity normalized canonical epithelial marker gene expression, averaged per subject, grouped by cell type, as shown in **a**. **d** Heatmap of differentially expressed genes in ABCs of IPF patients vs nonUIP ILD disease controls showing a distinct deviation of gene expression (each column is a ABC, each row a gene as z-scores), **e** Violin plots of a subset of differentially expressed genes (DEGs) split by disease state. **f** Bar plot of log10-transformed adjusted *p*-values from the pathway analysis using the human KEGG 2019 database.

cells[14,16], these cells differed in that they did not exhibit the senescence markers, CDKN2A and CDKN2B as well as MMP7, and expressed KRT5.

## ABCs from IPF patients generate significantly more bronchospheres than ABCs from healthy volunteers or nonUIP interstitial lung disease (ILD) patients in a 3D organoid model

ABCs were generated from airway epithelial cells obtained by bronchial brushings through outgrowth. We used BEGM to expand proliferating ABCs. After 21 days of cell culture in flasks, we received a cell population of highly enriched ABCs. More than 97% of the harvested cells at day 21 expressed KRT5 and KRT17 which mark them as ABCs. We did not observe any difference in either EPCAM, KRT5 or KRT17 expression of ABCs derived from IPF patients or healthy controls. While cell proliferation of IPF-ABCs compared to those obtained from healthy volunteers (HV-ABC) or individuals with fibrotic nonUIP ILD (NU-ABC) did not differ in 2D cell culture, the bronchosphere formative capacity in the 3D organoid model used by us was very different (Fig. 2a–d). All three cell types formed organoids in 3D culture after 21 days, however, IPF-ABC formed organoids that were multilayered and frequently contain hollow and tube-like structures, visually resembling bronchospheres (Fig. 2b, c) which consisted mainly of KRT5+ KRT6+ KRT17+ basal cells (Supplementary Fig. 3). The average number of spheres per well generated by IPF-ABCs was significantly higher compared to HV-ABCs or NU-ABCs ($120 \pm 73$, $24 \pm 26$, or $2 \pm 2$, respectively; $P < 0.0001$, Fig. 2d). In addition, cell proliferation as measured at day 21 by MTT assay was also highly increased in IPF-ABCs compared to HV-ABCs and NU-ABCs ($P < 0.0001$, Fig. 2e). Based on our scRNAseq data[13], we tested whether EGFR ligand production was upregulated in IPF-ABCs. Indeed, conditioned medium of 3D organoid cultures of IPF-ABCs showed higher levels of amphiregulin (AREG) than HV-ABCs or NU-ABCs ($P < 0.0001$), whereas HBEGF was not detectable. Bronchosphere counts correlated with amphiregulin levels at day 14 ($P < 0.0001$, $r^2 = 0.69$, Fig. 2f). In accordance with scRNAseq data, also CTGF production was highly increased in IPF-ABCs compared to NU-ABCs or HV-ABCs ($P < 0.0001$, Fig. 2g), while TGF-β was not detectable (data not shown).

## Presence of fibroblasts increased bronchosphere formation and IPF-ABCs stimulated fibroblast proliferation and collagen production

To assess the interaction of fibroblasts and ABCs we added primary lung fibroblasts to our 3D cell culture system. IPF-ABCs formed bronchospheres similar to the ones observed before, which were now surrounded by a mesh of fibroblasts (Fig. 2b). Presence of lung fibroblasts, particularly primary cells derived from IPF explants, in the 3D cell culture system, significantly enhanced bronchosphere formation of ABCs ($P = 0.012$ and $P = 0.001$, Fig. 2h, i). Using GFP-transduced fibroblasts, we found that the overall fluorescence was significantly increased by IPF-ABCs in healthy (HV-Fib) as well as IPF fibroblasts (IPF-Fib) ($P = 0.005$ and $P = 0.002$, respectively, Fig. 2j, k), potentially reflecting enhanced proliferation. IPF-ABCs significantly increased collagen production by lung fibroblasts compared to HV-ABCs ($P < 0.0001$, Fig. 2l). Stimulation of normal lung fibroblasts with conditioned medium obtained from NU-ABC or IPF-ABC harvested at day 14 induced fibronectin (Fig. 2m, n) and collagen (Fig. 2m, o) expression in lung fibroblasts as well as induced EGFR phosphorylation (Fig. 2p and Supplementary Fig. 5), a finding consistent with significantly increased secretion of amphiregulin by IPF-ABC bronchospheres (Fig. 2f).

## Human IPF-ABCs augment bleomycin induced pulmonary fibrosis and induce ultrastructural changes in $RAG2^{-/-}$ and $NRG$ mice

Intratracheal application of human IPF-ABCs or HV-ABCs to uninjured $Rag2^{-/-}$ mice and NRG mice had no discernible effect (data not shown). Thus, we decided to administer the human IPF-ABCs or HV-ABCs three days after causing the lung minimal injury with a low dose of bleomycin to $RAG2^{-/-}$ mice (1.2 mg/Kg body weight IT, Fig. 3a). The administered bleomycin dose resulted in mild fibrotic changes in the lungs (Fig. 3b). IPF-ABCs administration highly increased bleomycin induced pulmonary fibrosis compared to HV-ABCs or bleomycin alone as shown by Masson trichrome staining (Fig. 3b). Ashcroft score ($P < 0.0001$) and hydroxyproline levels ($P < 0.0001$) were significantly increased in mice challenged with IPF-ABCs compared to mice challenged with HV-ABCs or bleomycin alone (Fig. 3c, d). Similar results were obtained in NOD.Cg-Rag1tm1Mom Il2rgtm1Wjl/SzJZtm (NRG) mice (Supplementary Fig. 6). Intratracheal application of A549 or sorted ciliated cells from air–liquid interface cultures did not increase bleomycin induced fibrosis (Supplementary Fig. 6). Masson trichrome staining showed abundant new collagen production centered around the airway structures but also reaching up to the pleura (Fig. 3b). Starting with day 8 we observed formation of de novo airway structures, areas of bronchiolization and formation of cystic structures (Fig. 3e, f and Supplementary Fig. 7). Some of the induced lesions resemble pseudoglandular lesions described during lung development or recently described glandular-like epithelial invaginations[17] (Fig. 3e). As expected, engraftment and proliferation of human ABCs was enhanced in NRG mice. In NRG mice challenged with IPF-ABCs but not HV-ABCs we regularly observed focal squamous metaplastic lesions of human ABCs (Fig. 3g). In some experiments, NRG mice were challenged with human IPF-ABCs transfected with a luciferase and GFP encoding vector. Bioluminescence measurements of luciferase expression in mice injected with human IPF-ABCs (day 3–21) showed an increased signal intensity up to day 21 suggesting pulmonary engraftment of human cells and further proliferation (Fig. 3h). Introduction of GFP-transduced IPF-ABC to the lungs of NOD.Cg-Prkdcscid Il2rgtm1Wjl Tg(CAGGS-VENUS)1/Ztm (NSG) venus expressing mice confirmed the engraftment of these cells in the murine lung building up focal squamous metaplastic and pseudoglandular lesions (Fig. 3i and Supplementary Fig. 8). Using NSG mice we were able to observe engrafted IPF-ABCs adjacent to murine origin areas of pseudoglandular lesions/glandular-like epithelial invagination structures, supporting an effect of IPF-ABC cells on host resident cells.

## Single-cell sequencing of IPF-ABCs and NU-ABCs identifies SRC as a potential therapeutic target

Connectivity MAP analysis using the gene expression profile of IPF-ABC and NU-ABC identified a list of substance classes predicted to reverse the IPF-ABC signature. Three perturbagen classes had a maximum summary connectivity score of −100: SRC-inhibitors, MEK-inhibitors, and loss of function of C2 domain containing protein kinases (Fig. 4a). In addition, SRC-inhibition was the second best candidate with a summary connectivity score of −99.94 if we analyzed the ABC signature associated with mortality in a large cohort of IPF patients that we recently published[13].

## SRC expression is increased in lung tissues of IPF patients and murine lungs of the described humanized mouse model

Based on the results of the described in silico analyses, we analyzed SRC protein expression in IPF-tissues and ABCs. SRC was highly increased in IPF lung tissues, specifically in epithelial cells covering fibroblast foci and within areas of bronchiolization (Fig. 4b). SRC staining of macrophages was observed in both IPF and control lungs (Fig. 4c and Supplementary Fig. 9). SRC was increased in lung homogenates of IPF lung tissues compared to healthy lung tissue (Fig. 4d and Supplementary Fig. 10). Impressively, SRC protein expression was also observed in the xenograft IPF-ABC mouse model described above, primarily in areas of aberrant airway generation, bronchiolization and in glandular-like epithelial invagination lesions (Fig. 4e). As expected, SRC staining of macrophages was also observed in the mouse lung (Fig. 4e).

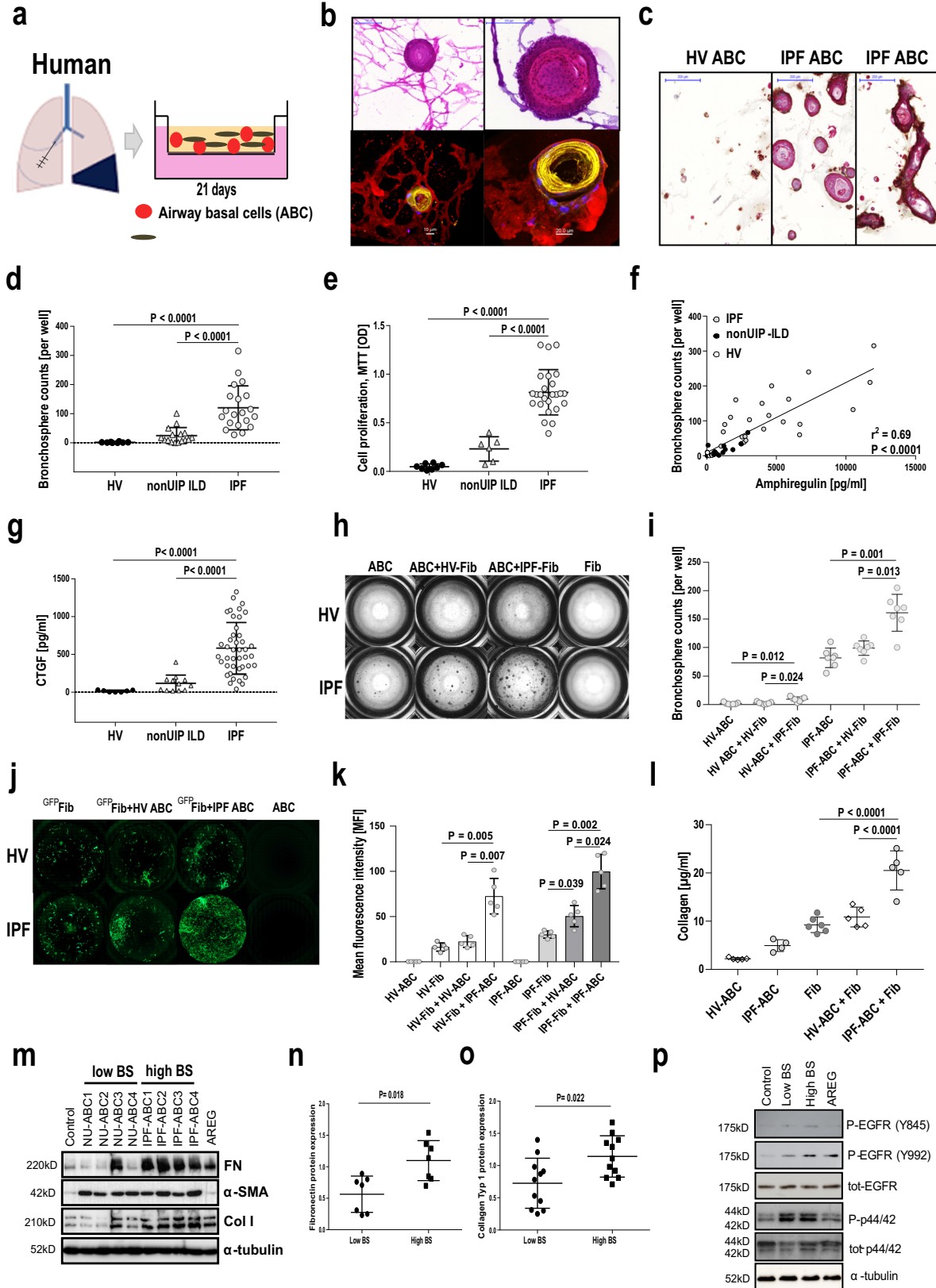

## SRC overexpression in IPF-ABCs increased cell invasion and fibrosis, while SRC knockdown attenuated fibrosis

IPF-ABCs were transduced with different lentiviral vectors, which lead to overexpression of SRC, knockdown of SRC or expression of GFP as control (empty vector (EV)). Effects on SRC expression were confirmed by Western blot (Fig. 4f and Supplementary Fig. 11). We tested the

effects of c-src expression levels in our 3D organoid model. IPF-ABCs, which overexpressed c-src, generated more spheres compared to IPF-ABCs transduced with an empty vector (Fig. 4g, h). Moreover, lentiviral knockdown of c-SRC expression let to reduced generation of bronchospheres (Fig. 4g, h). We then tested the effects of modulated SRC expression in IPF-ABCs on fibrosis in the IPF-ABC xenograft model

**Fig. 2 | IPF-ABCs generate more bronchospheres and increase fibroblast proliferation and collagen production compared to non-IPF ABCs in a 3D organoid model. a** Airway basal cells w/wo human lung fibroblasts were cultured in matrigel applying a transwell system. **b** IPF-ABCs form large spheres which become hollow tube-like structures after 21 days of 3D culture. In the co-culture system of ABCs with lung fibroblasts, fibroblasts surround bronchospheres and form a mesh-like structure. Upper panel: Masson trichrome staining, scale bars 100 μm or 200 μm as indicated. Lower panel: Confocal immunohistochemistry demonstrates tube formation by IPF-ABCs and close interaction with lung fibroblasts (red = vimentin, yellow = KRT5/6, blue = TO-PRO-3 = nuclei, $n = 10$, scale bar 10 μm and 20 μm). **c** Immuno-histochemistry of evolving bronchospheres stained for KRT5/6 in red, p40 in turquoise and beta6 integrin in brown ($n = 9$, scale bars 200 μm). **d** IPF-ABCs ($n = 23$) generated significantly more and larger spheres than HV-ABCs ($n = 7$; $P < 0.0001$) and NU-ABCs ($n = 15$, $P < 0.0001$); mean ± SD. **e** Cell proliferation was also significantly increased in IPF-ABCs ($n = 23$) compared to HV-ABCs ($n = 7$, $P < 0.0001$) and NU-ABCs ($n = 6$, $P < 0.0001$) as measured by MTT assay at d21 (mean ± SD). **f** Amphiregulin levels were increased in conditioned medium of bronchospheres from IPF-ABCs ($n = 23$) compared to HV-ABCs ($n = 7$) and NU-ABCs ($n = 15$) and correlated closely with bronchosphere counts. **g** CTGF levels (mean ± SD) were increased in conditioned medium of bronchospheres from IPF-ABCs ($n = 42$) compared to HV-ABCs ($n = 7$) and NU-ABCs ($n = 12$). **h** Bright field images of raster microscopy of an original experiment (10 independent experiments in triplicate). Lung fibroblasts do not form spheres. Sphere formation by IPF-ABCs is easily detectable. **i**, In the presence of lung fibroblasts ($n = 5$ IPF-Fib, $n = 5$ HV-Fib), IPF-ABCs and HV-ABCs generate increased numbers of bronchospheres (mean ± SD). **j** Fibroblast cell lines were transduced with lentiviral vectors encoding GFP. Fibroblast proliferation was highly increased in the presence of IPF-ABCs. **k** Mean fluorescence intensity was significantly increased in fibroblast cell lines co-cultured with IPF-ABCs ($n = 5$; mean ± SD). **l** Collagen levels were detected in conditioned medium and matrigel by sircol assay at day 63. Collagen production by lung fibroblasts cultured with IPF-ABCs was significantly increased ($n = 5$, mean ± SD). **m–o** in addition, normal lung fibroblasts cultured for 48 h in conditioned medium of IPF-ABC-derived bronchospheres (high BS) showed an increase in fibronectin and collagen 1A expression compared to conditioned medium of bronchosphere cultures derived from NU-ABCs (low BS). **n** Normalized expression levels of fibronectin from $n = 7$ fibroblast lines (mean ± SD). **o** Normalized expression levels of collagen-1 from $n = 11$ fibroblast lines (mean ± SD). **p** Normal lung fibroblasts were treated for 20 min with conditioned media of bronchospheres which were harvested at day 14. Pooled conditioned media of bronchospheres derived from IPF-ABCs (High BS) resulted in EGFR phosphorylation compared to conditioned media of bronchospheres derived from NU-ABCs (Low BS). For statistical comparison (**d**, **e**, **g**) one-way ANOVA with Tukey correction for multiple testing, **f** Pearson correlation, (**h**, **j**, **k**) repeated measures one-way ANOVA with Tukey correction for multiple testing, (**n**, **o**) two-tailed Mann–Whitney test was used.

described above. Forced overexpression of SRC in IPF-ABCs (SRC⁺ IPF-ABC) resulted in enhanced fibrosis and cellular remodeling in the alveolar compartment of *NRG* mouse lungs, whereas SRC knockdown in IPF-ABC cells (SRC⁻ IPF-ABC) from the same donor caused markedly reduced remodeling (Fig. 4i). Quantification of these results using the Ashcroft score confirmed both, a significant increase in fibrosis in SRC⁺ IPF-ABC and a decrease in fibrosis in SRC⁻ IPF-ABC compared to controls ($P = 0.023$, $P = 0.006$, respectively Fig. 4j).

## The SRC inhibitor saracatinib completely abrogates IPF-ABC bronchosphere formation and attenuates fibroblast proliferation in vitro

Based on the Connectivity MAP predictions, we tested whether saracatinib, a known SRC inhibitor previously tested in human cancer, is able to modulate IPF-ABC phenotype. Saracatinib treatment of IPF-ABCs for 24 h downregulated src phosphorylation at Tyr527 (Supplementary Fig. 12). When IPF-ABC were cultured in 3D and were treated with saracatinib, nintedanib, pirfenidone or vehicle, we observed a complete abrogation of bronchosphere formation at saracatinib concentrations of 600 nM, 210 nM, and 75 nM, whereas nintedanib and pirfenidone had no visible effect on bronchosphere formation (Fig. 5a). The results were similar in cells obtained from 12 different individuals with IPF ($P < 0.0001$, Fig. 5b). Saracatinib did not affect cellular vitality of ABCs in all used concentrations (Supplementary Fig. 13) and these concentrations were considered equivalent to clinically relevant doses[18]. Similar data were obtained by testing cell proliferation in the MTT assay ($P < 0.0001$, Fig. 5c). Saracatinib completely abrogated cell proliferation at concentrations of 600 nM and 210 nM. In the co-culture model of IPF-ABC with lung fibroblasts, saracatinib completely blocked bronchosphere formation at the concentrations of 600 nM, 210 nM, and 75 nM, had a lower effect at 25 nM, and did not have an effect at 8 nM (Fig. 5d–f). In contrast to the single culture model described above, in the IPF-ABC fibroblast co-culture model, pirfenidone and nintedanib significantly reduced bronchosphere formation, but not completely (Fig. 5d–f), potentially reflecting an effect of these drugs on the fibroblast component of this interaction. Fibroblast proliferation in the co-culture was also substantially lower in the presence of saracatinib (Fig. 5g, h).

## Saracatinib attenuated fibrosis and bronchialization in vivo

To test the effect of saracatinib on IPF-ABC induced fibrosis and remodeling in vivo, we returned to the minimal injury xenotransplant model in NRG mice. Overall, we used IPF-ABCs from 38 different individuals with IPF in these experiments. Per IPF subject, IPF-ABCs were used in two animals, one treated with saracatinib and one with vehicle control, to account for interindividual variability. To test the effect of saracatinib on engraftment and development of fibrosis we started treatment at day 4 post injury, 1-day post IPF-ABC installation for a total of 18 days (Fig. 6a). Oropharyngeal saracatinib in a dose of 10 mg/kg once daily significantly reduced fibrosis at day 21 as measured by Ashcroft score ($4.4 \pm 0.9$, $1.4 \pm 0.7$, $P < 0.001$, Fig. 6b) and hydroxyproline/total-protein levels ($2.4 \pm 1.0$, $0.7 \pm 0.6$, $P = 0.001$, Fig. 6c). To test the effect of saracatinib on established fibrosis we performed another set of experiments in which treatment was delayed to day 8 when cell engraftment and fibrosis were already established. Saracatinib treatment again significantly reduced remodeling and fibrosis (Fig. 6d) as was reflected by significant reductions in the Ashcroft score ($P = 0.004$, Fig. 6e) and hydroxyproline content ($P = 0.042$, Fig. 6f).

## Discussion

In this paper, we demonstrate that ABCs, which are found in areas of remodeling and bronchiolization and adjacent to fibroblastic foci in the IPF lung, have unique properties and are reprogrammed. Transcriptionally, IPF-ABCs are substantially different and exhibit enhanced stemness, ECM sensing and EGF signaling. Their KRT17^high PTEN^low phenotype is consistent with a cellular reprogramming during dedifferentiation. Using a 3D organoid model, we demonstrate that IPF-ABCs give rise to more bronchospheres compared to normal or non-UIP ILD cells in our conditions. Co-culture experiments with fibroblasts show a close interaction of both cell types which results in augmented bronchosphere formation by IPF-ABC as well as enhanced proliferation and extracellular matrix (ECM) deposition by fibroblasts. Intratracheal application of IPF-ABCs into minimally injured lungs of immunocompromised mice leads to severe fibrosis and remodeling of the alveolar compartment including the evolution of honeycomb cyst-like structures. Connectivity analysis suggested that gene expression changes in IPF-ABCs can be reversed by SRC inhibition and enhanced SRC expression and activity were observed in IPF lungs. Saracatinib, a potent SRC inhibitor, modulated the in vitro and in vivo IPF-ABC induced profibrotic changes.

In this paper, we provide the first demonstration that ABCs from patients with IPF are functionally different and have profibrotic effects in vivo and in vitro. ABCs are considered an airway stem cell population, capable of proliferation and self-renewal and can give rise to all types of airway epithelial cells[9]. In recent years, there have been

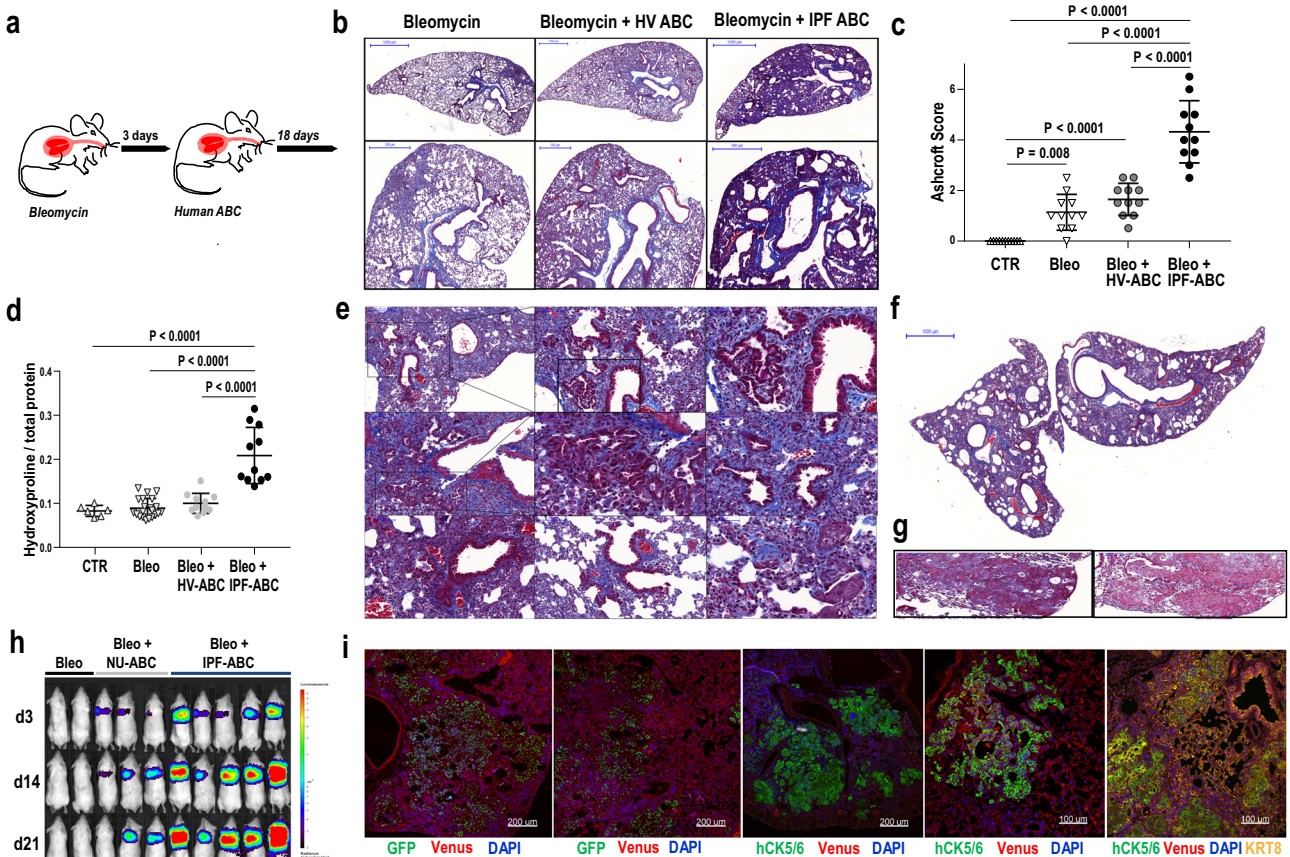

**Fig. 3 | Establishment of a humanized mouse model for IPF based on human ABCs. a** Bleomycin (Bleo) was intratracheally administered to *Rag2*$^{-/-}$ mice (BL6 background). Three days later human airway basal cells (ABC) derived from either patients with IPF or HV were intratracheally administered. Lungs were harvested at day 21 following Bleo application. **b** Representative trichrome stainings of mice challenged with Bleo alone, Bleo and HV-ABCs or Bleo and IPF-ABCs (each arm *n* = 11, 2 replicates with similar results). Fibrosis and cystic lesions are increased in mice challenged with Bleo + IPF-ABCs. **c, d** Ashcroft score (mean ± SD; 2 replicates) and hydroxyproline levels (mean ± SD; 2 replicates) were significantly increased in mice challenged with Bleo + IPF-ABCs (*n* = 11) compared to mice challenged with Bleo and HV-ABCs (*n* = 11) or Bleo alone (*n* = 21). **e, f** Trichrome staining of mice injected with Bleo + IPF-ABCs, shown are representative lesions of 11 different mice (*n* = 11, 2 replicates with similar results). Bronchiolization of the alveolar compartment and de novo generation of airway structures is highly increased in these mice compared to mice challenged with Bleo alone or Bleo + HV-ABCs. These bronchiolar lesions look often bizarre and undirected. Others appear to resemble honeycomb cysts. New collagen synthesis (shown in blue) is seen predominantly around the bronchiolar lesions. Rarely structures resembling fibroblast foci could be detected. **g** In some experiments with IPF-ABCs in NRG mice (*n* = 11, 2 replicates with similar results) metaplastic squamous lesions evolved. **h** Mice were challenged with human IPF-ABCs transfected with a luciferase and GFP encoding vector (*n* = 10, 3 replicates). Shown are representative bioluminescence measurements of luciferase expression in control NRG mice (Bleo) and NRG mice injected with transduced human IPF-ABCs and NU-ABCs (day 3–21). **i** Engraftment of human IPF-ABCs into venus expressing NSG lungs was detected by confocal microscopy (*n* = 6 with similar results). Human cytokeratin (hCK) 5/6, eGFP, CK8, nuclear DAPI, and Venus-expression were detected. Human IPF-ABCs generate focal metaplastic lesions and pseudoglandular lesions in murine lungs. Some of the pseudoglandular lesions are also derived from murine cytokeratin (CK)8$^+$airway epithelial cells. For statistical comparison (**c, d**) one-way ANOVA with Tukey correction for multiple testing was used. Scale bars: 50 μm (**e** center and third column, **i**), 100 μm (**e** center column top and bottom, left column 2nd and 3rd, **g, i** 4th and 5th figure), 200 μm (**e** top left, **i** figure 1 to 3), 500 μm (**a** second row), and 1000 μm (**b** first row, **f**).

cumulative evidence that in the IPF lung, ABCs migrate from their airway niche to the lung parenchyma, populating areas of honeycomb cysts and covering fibroblastic foci[6,7,13]. However, so far, it was unclear whether ABCs were acting as active players in lung remodeling or whether they served as merely innocent bystanders. We provide several lines of evidence that IPF-ABCs have substantial profibrotic properties and are dedifferentiated. The first clue comes from the observation that IPF-ABCs—unlike HV-ABCs or NU-ABCs—generate numerous, well-developed and multilayered bronchospheres in the 3D culture model system we used, as well bronchial-like structures in vitro, both suggestive for exaggerated stem cell phenotype and dedifferentiation of these cells.

This is also supported by the scRNAseq results, as IPF-ABCs overexpress known stemness genes such as transcription factors and inducers of pluripotency KLF4 and MYC, as well as AP1 forming c-JUN and FOSL1, which are downstream transcription factors of genes inducing pluripotency[19] and important oncogenes for cancer stem cells[20]. Moreover, PTEN expression was highly decreased in IPF-ABCs and decreased PTEN expression is associated with increased PI3K signaling and increased stem cell renewal. Of interest, the signature of KRT17$^{high}$IPF-ABCs appears to be similar to the recently published signature of KRT17$^+$ SOX4$^{high}$ aberrant basaloid cells which lack KRT5[14–16].

The second line comes from the co-culture experiments that suggested a self-amplifying interaction between ABCs and fibroblasts. The presence of lung fibroblasts increased evolution of bronchospheres, whereas ABCs stimulated proliferation and collagen production of fibroblasts. In both cases, the effect was stronger when IPF cells were used. Of note, TGF-β was not produced by IPF-ABCs. Our scRNAseq data suggested that one profibrotic factor released by IPF-ABCs is CTGF and indeed IPF-ABCs produced significantly more CTGF than HV-ABCs or NU-ABCs. Overexpression of CTGF leads to pulmonary fibrosis in mice and elevated expression levels were reported

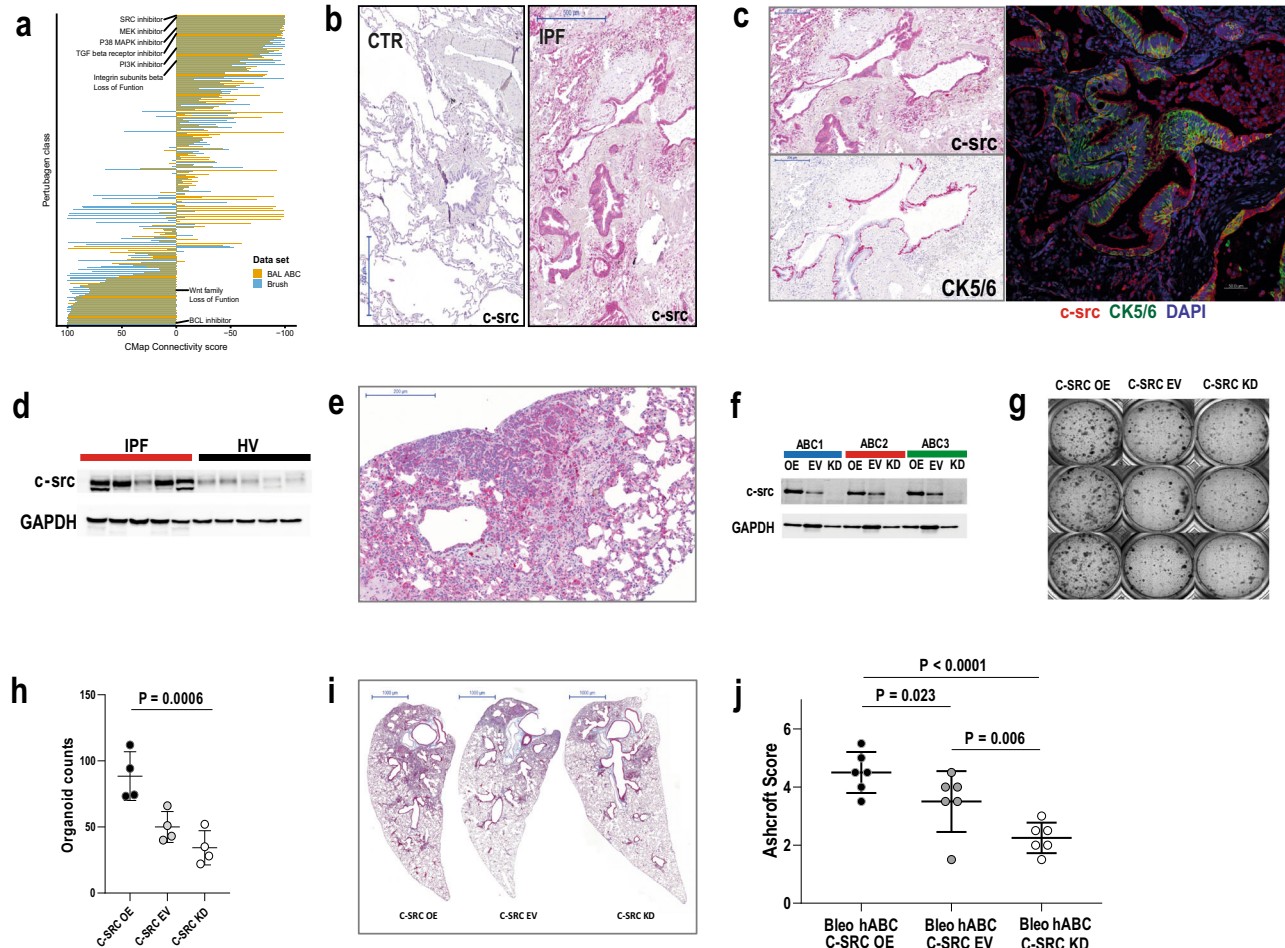

**Fig. 4 | CMap of scRNAseq data identifies potential treatment strategies such as src inhibition.** Overexpression of c-src in IPF lungs and cellular effects of c-src overexpression and knockdown. **a** Bar plots of summary CMap Connectivity scores of the clue.io analysis, dodged by input (ABC mortality signature—yellow, scSeq-derived DEGs IPF vs nonUIP ILD control in ABCs—blue). Inhibition of SRC had the lowest CMap Connectivity score, i.e., potentially reversing the input gene signatures. **b, c** In normal human lung tissue ($n = 5$) c-src expression was low, only few macrophages were stained positively. In contrast c-src expression of IPF lung tissue ($n = 10$) was high. **b** Macrophages and epithelial cells highly expressed c-src as well as lymphocytes within lymphoid follicles. **c** Most of the epithelial cells expressing c-src were also positive for CK5/6 identifying them as basal cells. **d** c-src expression was high in lung tissue homogenates of IPF patients compared to healthy donors. **e** Immunohistochemistry of murine lungs ($n = 10$) of the described humanized

mouse model also showed high c-src expression of macrophages and airway epithelial cells. **f** c-src expression was either overexpressed, untouched (EV), or knockdown in human ABCs using a lentiviral vector and resulting c-src expression was measured using Western blot ($n = 12$ with similar results). **g, h** 3D organoid cultures of transduced IPF-ABCs were prepared. IPF-ABCs overexpressing c-src generated more spheres and downregulation of c-src expression let to reduced sphere formation (mean ± SD, $n = 4$, 3 replicates with similar results). **i, j** c-src overexpression in human IPF-ABCs which were injected into *NRG* mice lead to increased pulmonary fibrosis, while knockdown of c-src downregulated the induced fibrosis (mean ± SD, $n = 6$, each group, 2 replicates with similar results). For statistical comparison repeated measures one-way ANOVA with Tukey correction for multiple testing was used. Scale bars: 500 μm (**b**), 200 μm (**c**), 200 μm (**e**), 1000 μm (**g**).

in humans with fibrotic lung disease[21,22]. Another contributing mechanism to this phenomenon involves the EGFR axis: scRNAseq revealed that amphiregulin expression is significantly increased in IPF-ABCs compared to NU-ABCs and that the bronchospheres generated from IPF-ABCs secrete significantly higher concentrations of amphiregulin compared to both HV-ABCs and NU-ABCs. Conditioned media from IPF-ABC bronchosphere cultures induced phosphorylation of EGFR in fibroblasts. Generally, the role of EGFR ligand family members in fibrosis has been studied mainly in other organs with sometimes conflicting results[23]. In the lung, amphiregulin was proposed as a mediator of TGFB1 mediated pulmonary fibrosis in the triple transgenic mouse model[24], but without significant follow-up. Our results suggest that amphiregulin may be a frequently overlooked major mediator of the interaction of ABCs and fibroblasts in fibrosis.

The third line of evidence comes from the humanized model of lung fibrosis we established. Human IPF-ABCs, but not HV-ABCs, induced abundant bronchiolization, airway enlargement, cyst

formation, and frequently pleural extending fibrosis, hallmark features of UIP histology missing in commonly used animal models of pulmonary fibrosis. Taken together these lines of evidence support the unique profibrotic properties of ABCs obtained from lungs of patient with IPF.

The reason why IPF-ABCs are different is unclear, but they are clearly very different from NU-ABCs and show a dedifferentiated phenotype. Recent publication call this phenotype hyper- and metaplastic and indeed did we find metaplastic lesions of human basal cells in our mouse model. We speculate that cigarette smoking, exposure to other environmental factors, recurrent airway infections and genetic background may lead to repetitive airway epithelial barrier injury and a chronic wound healing response in IPF, which then may have an impact on ABC gene expression as recently described for asthma and TH2 inflammation[25]. Genetic risk factor for IPF such as polymorphisms in MUC5B, desmoplakin, and AKAP13 are linked to epithelial barrier function and AKAP13 and desmoplakin, unlike MUC5B, are highly

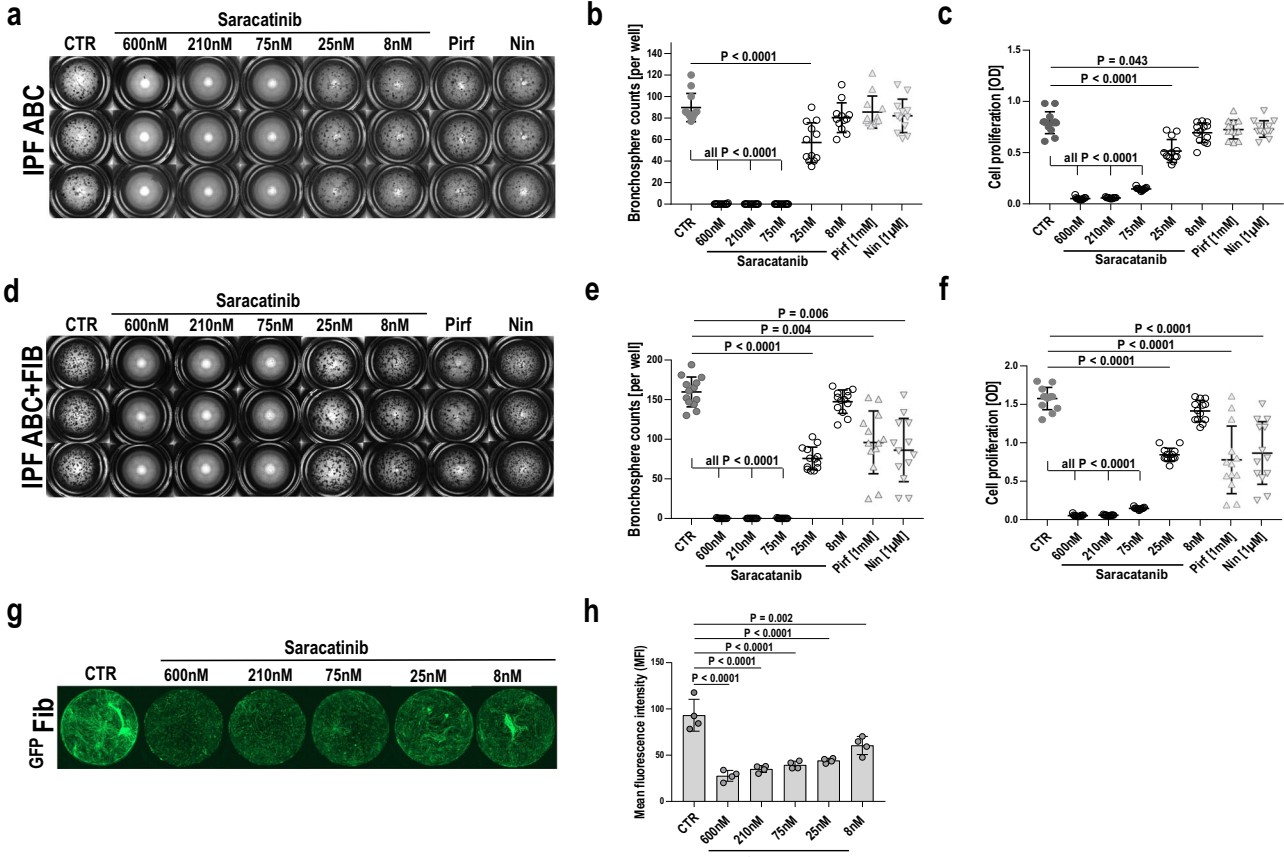

**Fig. 5 | SRC-inhibitor saracatinib abrogates bronchosphere formation.**
**a–f** Bronchosphere assays of human IPF-ABCs were performed w/wo IPF lung fibroblasts for 21 days. Cell cultures ($n = 12$, in triplicates) were treated with either vehicle, saracatinib (600, 210, 75, 25, 8 nM), pirfenidone (1 mM) or nintedanib (1 μM). **b** Saracatinib abrogated sphere formation dose-dependently while a dose of pirfenidone or nintedanib, which was higher than usually applied in humans, did not change bronchosphere counts (mean ± SD). **c** Cell proliferation as measured by MTT assay was significantly reduced by saracatinib treatment in a dose-dependent manner and slightly by pirfenidone and nintedanib (mean ± SD). **d, e** Saracatinib treatment also abrogated sphere formation dose-dependently in the presence of

IPF lung fibroblasts while pirfenidone and nintedanib reduced significantly, but not abrogated, sphere formation in the presence of fibroblasts (mean ± SD). **f** Cell proliferation as measured by the MTT assay was significantly reduced by saracatinib treatment in a dose-dependent manner and less reduced by nintedanib or pirfenidone treatment (mean ± SD). **g, h** In the same model we used GFP+fibroblasts to study fibroblast proliferation. Saracatinib treatment also showed a dose-dependent effect upon fibroblast proliferation (mean ± SD, $n = 4$, 2 replicates). For statistical comparison repeated measures one-way ANOVA with Tukey correction for multiple testing was used.

expressed by IPF-ABCs[26]. Our scRNAseq data demonstrate increased expression of the stress-induced keratins KRT6 and KRT17[27], upregulation of key molecules of epithelial barrier function (claudin 1 and 4), of several integrins which regulate ECM composition (β6, vα, β1, α6, and α2), markers of epithelial cells senescence such as GDF15[28], and the EGFR ligands AREG and HBEGF in IPF-ABCs. Thus, the transcriptional phenotype of IPF-ABCs may represent the end results of the lung response to repetitive epithelial injuries[29], which trigger exaggerated repair processes including dedifferentiation and epithelial-mesenchymal crosstalk[6,30].

Another feature of our study is the focus on SRC inhibition in pulmonary fibrosis and especially as mediator of IPF-ABC profibrotic effects. Based on our scRNAseq data and our recently published BAL dataset[13] we queried for potentially beneficial drug candidates using connectivity map. Our in silico analyses indicated that SRC-inhibitors may be capable of reversing the profibrotic IPF-ABC phenotype. SRC is a hub integrating multiple pathways including integrin and receptor kinase signaling[31], and can phosphorylate various substrates such as STAT3, FAK, JNK, EGR, AKT, and PI3K. Allergic inflammation in murine asthma model is reported to be regulated by SRC trans-activation of EGFR, followed by activation of ERK1/2/NFκB[32]. Mechanosensitive activation of Beta1 integrin and the downstream

signaling via focal adhesion kinase (FAK) and SRC are increased in multiple cancers and linked to proliferation, migration, and invasion of cells[31,33]. In human basal layer keratinocytes, SRC is known to induce, together with integrins and PI3K signaling, nuclear localization of YAP, therefore promoting cell proliferation[34]. SRC was also shown to align to the EGFR molecule to facilitate it's signaling both upstream and downstream and is linked to stemness of cancer cells[31,35,36]. SRC signaling is closely linked to epithelial injury[37–39] induced by various noxious agents including cigarette smoking[40]. A role for SRC has been proposed in fibrosis in multiple organs but most of this work focused on the role of SRC in fibroblasts[41–43]. We found abundant SRC expression in ABCs covering the fibroblast foci in IPF lungs. Overexpression of SRC in ABCs resulted in an increase in fibrosis and knockdown of SRC reduced fibrosis in our humanized mouse model of IPF. We tested the SRC and ABL inhibitor saracatinib (AZD0530)[44,45] in our in vitro and in vivo models. Saracatinib completely abrogated sphere formation in the 3D organoid model. In the co-culture model, saracatinib had an effect on both sphere formation and fibroblast proliferation suggesting that combined SRC and ABL inhibition by saracatinib may disrupt the profibrotic HV-ABCs crosstalk between IPF-ABCs and fibroblasts. Similar effects were observed in our humanized mouse model, where saracatinib

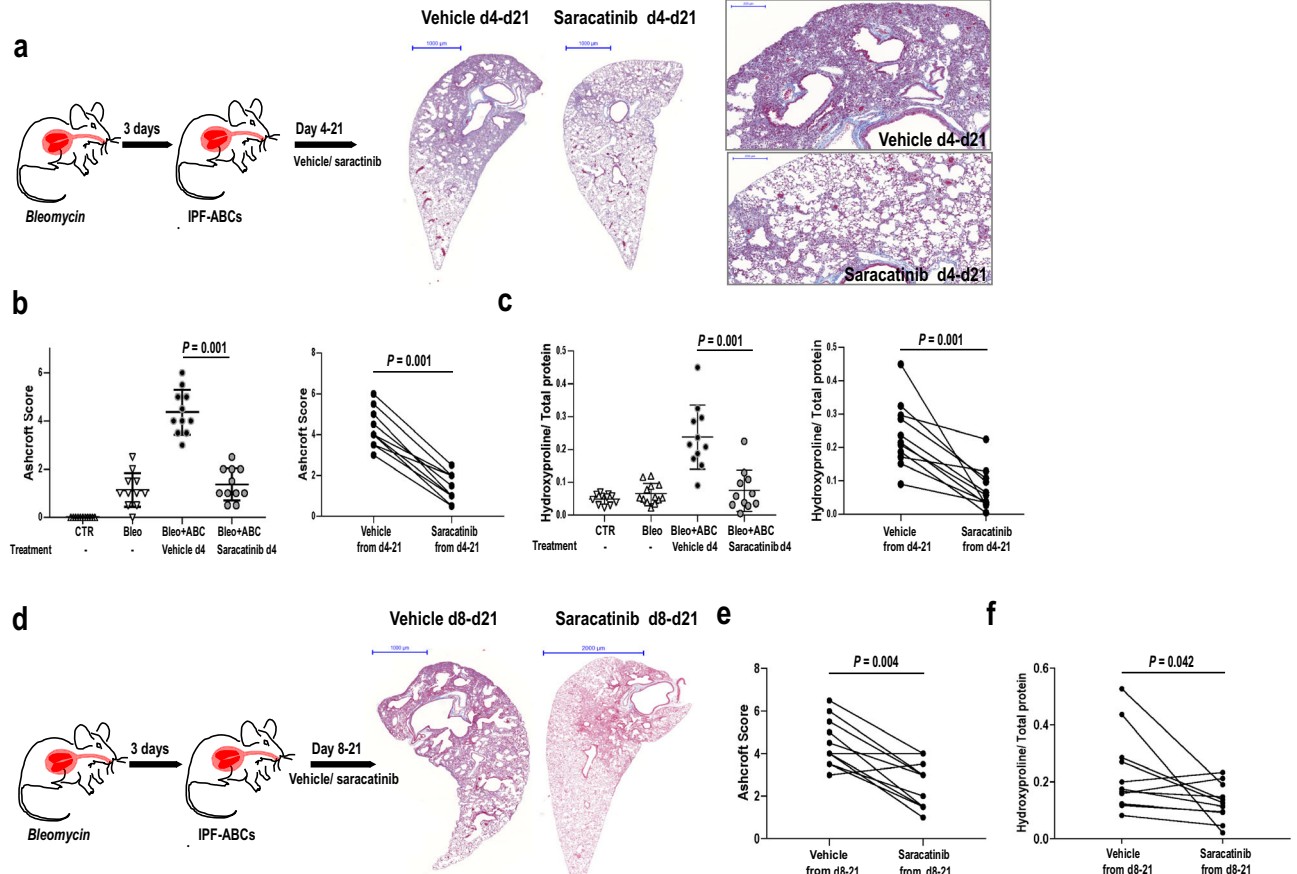

**Fig. 6 | SRC-inhibitor saracatinib attenuates fibrosis in the humanized mouse model.** *NRG* mice were intratracheally injected with a low dose of bleomycin and 3 days later with human ABCs derived from IPF patients. Mice (*n* = 11 each group, 4 independent experiments (all data shown)) were treated oropharyngeally with or without saracatinib in a dose of 10 mg/kg once daily starting at day 4 (**a**–**c**) or at day 8 (**d**–**f**). **a**–**c** Treatment with saracatinib from day 4 significantly reduced the evolution of fibrosis as measured by Ashcroft score (**b**, mean ± SD) and hydroxyproline

levels (**c**, mean ± SD). **d**–**f** Saracatinib treatment from day 8 to day 21 also significantly reduced fibrosis but effect was less than with immediate treatment as measured by Ashcroft score (**e**) and hydroxyproline levels (**f**). All *P* values were determined by a paired comparison (two-tailed Wilcoxon matched pairs signed rank test) testing the effect of saracatinib treatment for each human IPF cell line. Scale bars: 200 μm (**a**), 1000 μm (**a**, **d**), and 2000 μm (**d**).

substantially attenuated IPF-ABC-induced pulmonary fibrosis, bronchiolization, and cyst formation. This effect was also observed when saracatinib was administered during established fibrosis. Moreover, in a parallel publication we reported an anti-fibrotic effect of saracatinib on isolated human fibroblasts and within the bleomycin model, if used therapeutically[46]. The effect of saracatinib may not be soley attributed to src inhibition since the compound also inhibits the ABL kinase which is expressed in the lung epithelium and may be involved in lung repair after injury[47]. On the background of our data and previous publication a clinical trial testing the efficacy of saracatinib in IPF patients was initiated (NCT04598919). Taken together, these findings suggest that saracatinib treatment efficiently blocked human IPF-ABC engraftment and proliferation.

In conclusion, our data indicate a dedifferentiated phenotype and profibrotic role of ABCs in IPF. In recent years, there has been a significant increase in the understanding of the mechanisms of pulmonary fibrosis[1,2]. Roles for distinct lung cell populations such as fibroblasts and myofibroblasts, macrophages, and even alveolar epithelial cells have been proposed. While ABCs have been noticed in the IPF lung, mechanistic studies evaluating their properties and role in pulmonary fibrosis have been missing. Our results provide clear evidence that human IPF-ABCs are dedifferentiated and have profibrotic properties in vitro and in vivo, and that interventions that address their properties reverse fibrosis. These findings position the ABC as a key

cell in the pathogenesis of human pulmonary fibrosis and thus a novel cellular target for therapeutic interventions.

## Methods
### Experimental model and subject details
**Human specimens.** All experiments with human tissue samples were performed under protocols approved by the Institutional Review Boards at Hannover Medical School (approved IRB protocols #2518-2014 and #2923-2015) and University Medical Center Freiburg (approved IRB protocols #239/12 and #03/10). All patients and healthy volunteers signed informed consent prior to inclusion to the study. In total, bronchial brushes from 68 patients with idiopathic pulmonary fibrosis (IPF), 25 patients with nonUIP fibrotic ILD and 18 healthy volunteers of an older age (>50 years) obtained. Only patients with an idiopathic UIP and HRCT consistent with a "definite" UIP pattern were included. We did not notice any sex-dependent effects. In addition, lung explant tissues of in total 20 patients with IPF, 10 patients with COPD and normal tissue from 5 donor lungs were used for fibroblast isolation, lung homogenates and immunohistochemistry. Bronchial brushes of 6 patients with fibrotic interstitial lung disease, who did not have a UIP pattern on high-resolution computer tomography (HRCT) and bronchial brushes of 9 patients with IPF and a definite UIP pattern on HRCT were obtained for single-cell RNA sequencing (for further details see Supplementary Table 1). IPF diagnosis and other ILD

diagnosis was established by a multidisciplinary board according to the American Thoracic Society/European Respiratory Society criteria[48] and was later determined to be consistent with recent guidelines[3,49]. Samples for the bronchosphere formation assays were obtained from IPF patients ($n = 42$, 7 females, $69 \pm 10$ years of age), nonUIP patients ($n = 15$, 3 females, $65 \pm 11$ years of age) and healthy donors ($n = 7$, 1 female, $60 \pm 7$ years). No age- or sex-related statistical differences were observed in the described experiments.

**Mice.** All mouse procedures were conducted in accordance with the German law for animal protection and the European Directive 2010/63/EU and were approved by the respective local government (Regierungspräsidium Freiburg, Germany; AZ: 35-9185.81/G-14/17) and Lower Saxony State Office for Consumer Protection and Food Safety in Oldenburg/Germany (LAVES); AZ: 33.12-42502-04-15/1896 and AZ: 33.19-42502-04-15/2017), AZ:33.19-42502-04-17/2612). Different mice strains were used for the experiments as indicated. B6;129Sv-Rag2tm1Fwa/ZTM[50] (*Rag2$^{-/-}$*) and NOD.Cg-Rag1tm1Mom Il2rgtm1Wjl/SzJZtm[51] (*NRG*) were obtained from the central animal facility (Hannover Medical School, Hannover). NOD.Cg-Prkdcscid Il2rgtm1Wjl Tg(CAGGS-VENUS)1/Ztm (*Venus-NSG*) mice were kindly provided by Dr. Wiebke Garrels (Hannover Medical School). *Venus-NSG* were generated by cytoplasmic plasmid injection of a non-autonomous transposon system[52,53]. All cells of this mice stock are expressing a native live imaging fluorescent protein Venus[54]. Mice were housed at the animal unit of the University Medical Center Freiburg or Hannover Medical School (Central Animal Facility). At both institutions, mice were housed in specific pathogen-free (SPF) conditions in accordance with institutional guidelines and ethical regulations. All animals used in this study had no previous history of experimentation and were naive at the time of analysis. *RAG2$^{-/-}$* mice used in this study were maintained on the C57BL/6J background for >6 generations. All *RAG2$^{-/-}$*, *NRG*, and *Venus-NSG* mice used in this study were male and 8–12 weeks old.

### Method details

**Bronchoscopy.** Bronchial epithelial cells were harvested by bronchial brushes of sub-segmental bronchi of the right lower lobe during flexible bronchoscopy. In patients, this was done within the routine diagnostic work-up at initial diagnosis. None of the patients received anti-fibrotic treatment prior to bronchoscopy. Healthy volunteers underwent bronchoscopy voluntarily for study purposes. None of the included subjects was currently smoking.

**Isolation of airway basal cells (ABCs).** ABCs were isolated from bronchial brushes of sub-segmental bronchi of the right lower lobe using a similar protocol as described[13,55]. Bronchial brushes were placed in 2 ml of pre-warmed (37 °C) Clonetics™ Bronchial Epithelium Cell Growth Medium (BEGM) (Lonza, #CC-3170). Cells were detached from the brush by flicking the brush. Then, airway epithelial cells were pelleted by centrifugation ($250 \times g$, 5 min) and disaggregated by resuspension in trypsin/EDTA-solution (0.05%/0.02%) (Merck, #L2143) for 5 min at 37 °C. Trypsinization was stopped by addition of HEPES buffered saline (Merck, #391340) supplemented with 15% fetal bovine serum (Merck, #TMS-016-B), then cells were again pelleted at $250 \times g$, 5 min. The pellet was washed once with 5 ml of phosphate-buffered saline (PBS, Thermo Fisher, #14190250). Afterward, the cell pellet was resuspended in 5 ml of BEGM and seeded in T25 flasks (Merck, #CLS3056) in BEGM, supplemented with growth factors according to the manufacturer's instructions. The antibiotics supplied by the manufacturer of BEGM were used and additionally 0.1% amphotericin B (BioWhittaker Amphotericin B Antifungal Lonza, #17-836 R) and 1% penicillin-streptomycin (10,000 U/ml), (Merck, #A 2212) were added. Cultures were maintained in a humidified atmosphere of 5% $CO_2$ at 37 °C. Unattached cells were removed by changing of media after 18 h. Thereafter, media was changed every 7 days and cells were harvested

at day 21, when the cells were 90% confluent. Therefore, cells were trypsinized, harvested, and counted. Purity of the ABC population was determined by immunocytology (KRT5 staining) of cytospins and always exceeded 97%.

**Purity control by flow cytometry.** Human ABCs were trypsinized, harvested, and counted as described above. Then, 500,000 cells were washed with 500 μL PBS (ThermoFischer Scientific, #14190250), centrifuged at $250 \times g$, and resuspended in 50 μl cell staining buffer, containing 0.5% BSA (Sigma-Aldrich, #SLBX7626) and 2 mM EDTA (Sigma-Aldrich, #798681). To prevent non-specific antibody binding, cells were blocked with 2 μl FcR Human Blocking Reagent (Miltenyi Biotec, #130-059-901) for 20 min at 4 °C. Staining of the EpCAM was performed directly after blocking of the cells by incubating the cells with antibody for 20 min at 4 °C. Prior staining of cytokeratins for 20 min at 4 °C, cells were fixed and permeabilized with BD Cytofix/Cytoperm™ Kit (BD Biosciences, #554714) as recommended by the manufacturer. After two washing steps, cells were resuspended in 150 μl staining buffer and underwent flow cytometry. Data acquisition was conducted using CytoFLEX S Flow Cytometer (Beckman Coulter). Compensation control was performed with UltraComp eBeads™ Plus Compensation Beads (Invitrogen™, # 01-3333-41). Data analysis was performed using FlowJo Software v10.6.1 (BD Life Sciences), whereby cell debris and cell doublets were excluded from the analysis. Purity of the ABC population was determined by flow cytometry (KRT5 staining) and always exceeded 97%. All EPCAM$^+$KRT5$^+$ ABCs also expressed KRT17 (data not shown). All primary antibodies that were used for flow cytometry are listed in Supplementary Table 2.

**Isolation of primary fibroblasts.** Fibroblasts were isolated from fresh tissue blocks (0.3 cm × 0.3 cm × 0.3 cm) derived either from IPF lung explants ($n = 10$) or healthy donor lung ($n = 5$), using a protocol previously described by us[56]. In brief, small blocks of lung tissues were washed three times in PBS at 4 °C. Tissue blocks were then cultured in Gibco DMEM (Dulbecco's modified Eagle medium, Thermo Fisher Scientific, #11995-065) supplemented with 10% FBS, 1% penicillin/streptomycin and 0.1% amphotericin B in 6 well plates and resulted in the outgrowth of fibroblasts. Blocks were removed after 12 days. Outgrown fibroblasts were harvested at day 21 when cells reached confluence. Afterward, fibroblasts were trypsinized for 3 min at 37 °C, then immediately suspended in DMEM supplemented with 10% FBS, then pelleted at $250 \times g$ for 5 min, and seeded in T75 flasks (Merck, #CLS3276). Fibroblasts were cultured in DMEM with 10% FBS for 10 days until cells reached confluence, then harvested using the same trypsinization protocol as described above. Fibroblasts were archived in liquid nitrogen in aliquots of $1 \times 10^6$ cells per vial in 20% dimethyl sulfoxide (Merck, #276855) + 80% FBS containing DMEM freezing medium for subsequent experiments.

**Bronchosphere assay (3D organoids).** In order to test sphere formation and evolution of 3D organoids, a protocol described for tumor sphere formation was used[57]. Human ABCs ($10^4$ cells) and/or primary human lung fibroblasts ($10^4$ cells) were added to 50 μl ice-cold matrigel (corning® matrigel® matrix (Corning®, #356231) in transwell inserts (Corning Lifesciences Costar, #3470) and cultured for 30 min in the incubator (5% $CO_2$, 37 °C) until the matrigel became stiff. Then 600 μl of a 1:1 ratio of BEGM (Lonza, Basel, Switzerland, #CC-3170) and DMEM (Dulbecco's modified Eagle medium (Gibco, Swit Fisher Scientific/Germany) was added below the insert and additional 100 μl on top of the insert. Plates (Corning Lifesciences Costar, #3470) were cultured at 5% $CO_2$, 37 °C as indicated. Medium was exchanged every 7 days and conditioned media were stored at −80 °C. Conditioned medium of the 3D organoid culture (day 7 and day 14) was used for ELISA and fibroblast stimulation experiments. Sircol assay (Scientific-Biocolor/UK,

#S1000) was performed as recommended by the manufacturer on matrigel and conditioned medium harvested after 6 weeks of cell culture. Mosaic photomicrographs were taken from 3D bronchospheres w/o fibroblasts and from GFP expressing primary fibroblasts by bright field and fluorescence microscopy using Axio Observer Inverted microscope/Zeiss® and processed by ZEN microscope navigation Software (version 2.5 (blue edition)). Fibroblast proliferation was quantified by mean fluorescence intensity (MFI) which was measured by ImageJ 1.53Java1.8.0_331[64-bit] Software as recently described[58]. For histology, immunohistology, and confocal laser microscopy 3D cell cultures were cryopreserved using Tissue-Tek and cryomolds (Hartenstein/Germany; #CMM).

**Air–liquid interface culture (ALI).** Human ABCs were cultured in PneumaCult™-Ex Plus Medium (STEMCELL Technologies) in T25 flasks, pre-coated with collagen I (Corning), at 37 °C, 95% $O_2$, and 5% $CO_2$. After two weeks of culturing, cells were seeded onto Corning® Transwell® Cell Culture Plate (Corning), pre-coated with collagen IV (Sigma-Aldrich), at a density of 30,000 cells per well. In the first, expansion phase, 100 µl of PneumaCult™-Ex Plus Medium (STEMCELL Technologies) were added to the cells and 600 µl were added to the bottom chamber. After reaching confluency (7–10 days), the medium was aspirated from the upper chamber and the medium from the bottom chamber was replaced by PneumaCult™-ALI-S Medium (STEMCELL Technologies), to promote the second, differentiation phase, characterized by maturation of pseudostratified mucociliary epithelium. Medium change was performed every other day. Experiments were performed with ALI-cultures after ciliar beating was visually detectable (~4 weeks after ALI). Cells were then harvested and stained for EPCAM and PROM1. PROM1$^+$ cells were sorted by flow cytometry (Becton-Dickinson FACSAria III Fusion).

**Treatment of 3D bronchospheres with saracatinib, nintedanib, and pirfenidone.** 3D bronchospheres were cultivated in a transwell system in matrigel w/wo treatment with vehicle, saracatinib (in a concentration of 600 nM, 210 nM, 75 nM, 25 nM, or 8 nM; kindly provided by AstraZeneca)), pirfenidone (1 mM, Santa Cruz Biotechnologie, #sc-203663), nintedanib (1 µM, BIPF 1120, Santa Cruz Biotechnologie, sc-364433), or PBS (control). Medium was exchanged every 7 days.

**Detection of bronchosphere counts and measurement of cell proliferation.** Numbers of bronchospheres per well were counted by bright field microscopy on day 21. Only spheres with a size of 5 µm or larger were counted. Cell proliferation was quantified using the colorimetric MTT assay (Sigma-Aldrich, #CT01) on day 21. Briefly, 70 µl of MTT (3-(4,5-dimethylthiazol-2-yl)-2,5-diphenyl tetrazolium bromide) were added into the medium covering the organoids, gently resuspended, and then incubated at 37 °C for 4 h. Then 100 µl isopropanol in 0.04 N HCL were added and the well plate was placed on an orbital shaker (GENEO BioTechProducts GmbH, #1120-221-01) at 150 rpm. Isopropanol dissolves the formazan crystals to a homogenous blue solution. Optical density (OD) was determined at a wavelength of 570 nm using the Infinite 200 PRO Tecan microplate reader (Tecan Group Ltd., INF-FPLEX).

**Influence of c-src expression of human IPF-ABCs on bronchosphere generation.** 3D bronchospheres were generated from human IPF-ABCs which were transduced with lentiviral vectors to either overexpress c-src (OE), knockdown c-src (KD) or leave c-src untouched (empty vector, EV) as described above. Numbers of bronchospheres per well were counted by brightfield microscopy.

**Calcein AM/ethidium homodimer-1 ("LIVE/DEAD®") staining of 3D bronchospheres.** Viability of cells in the 3D bronchospheres was tested by fluorescence microscopy at day 1, 3, 7, 14, and 21 using the

Live/Death Staining kit. 3D organoids w/wo fibroblasts were incubated with 2 µM Calcein AM (Thermo Fisher, #C1430) and 5 µM Ethidium homodimer-I (EthD-1) (Invitrogen, Thermo Fisher, #E1169) for 45 min at RT in the dark on an orbital shaker at 150 rpm as recently described[59]. Dye solution was removed and bronchospheres w/wo fibroblasts were washed three times with 300 µl PBS by shaking at 150 rpm for 5 min at RT in the dark on an orbital shaker. Mosaic photomicrographs were taken by fluorescence microscopy using Axio Observer Inverted microscope/Zeiss and ZEN microscope navigation Software. Calcein was detected in the Cy2 channel and Ethidium homodimer in the Cy5 channel.

**Cryopreservation of 3D bronchospheres.** For histology, immunohistology and confocal laser microscopy, 3D organoids were cryopreserved using Tissue Tek O.C.T.™ compound (Hartenstein, #TTEK). The bottom of the transwell insert were excised with a scalpel and embedded in 2 ml Tissue Tek O.C.T.™ compound and placed in a cryomold (Hartenstein, #CMM). Cryomolds containing the 3D organoids were incubated on dry ice for 15 min. Cryopreserved bronchospheres were stored at −80 °C. Cryosections were prepared using the cryotome Leica CM 1900 (Leica Biosystems Cryostats).

**Measurement of collagen production by Sircol™ Collagen Assay.** Conditioned medium and conditioned matrigel of sphere cultures were harvested after 63 days and stored at −80 °C. The conditioned media and matrigels of 3D bronchosphere cultures were used for Sircol assay (Sircol™ Collagen Assay, Biocolor), which was performed as recommended by the manufacturer and recently described[59]. Briefly, conditioned media and matrigels were digested using the isolation and concentration reagent (polyethylene glycol in a TRIS-HCL buffer pH 7.6) in low binding protein tubes (Thermo Fisher, #90410) to release collagen at 4 °C overnight. 1 ml Sircol Dye Reagent was added and incubated on an orbital shaker at 150 rpm for 30 min. During this time period collagen-dye complexes were formed and precipitated out. Afterward, the tubes were centrifuged and washed with ice-cold Acid Salt Wash Reagent (acetic acid, sodium chloride, and surfactants) once. Then the collagen dye complexes were dried at RT. Finally, collagen-bound dye was dissolved by alkali reagent (0.5 M sodium hydroxide) and optical density (OD) of samples was determined using the Infinite 200 PRO Tecan microplate reader at a wavelength of 555 nm.

**Amphiregulin and CTGF detection by enzyme-linked immunosorbent assay (ELISA).** Conditioned media of 3D bronchosphere cultures were harvested after 14 days of cell culture and stored at −80 °C. Amphiregulin levels of the conditioned media were determined by ELISA following the manufacturer's instructions (R&D Duosets, #DY262). CTGF levels of the conditioned media were determined by ELISA following the manufacturer's instructions (R&D Duosets, #DY9190-05). OD was determined at 450 nm (reference wavelength 540 nm) using OKa Tecan reader Infinite 200 PRO (Crailsheim, Germany).

**Single-cell RNA sequencing of bronchial brushes.** Single-cell RNA sequencing (scRNAseq) was performed on bronchial epithelial cells derived from nine patients with IPF and six patients with nonUIP fibrotic ILDs. For baseline characteristics of IPF patients and nonUIP ILD patients, see Supplementary Table 1. For scRNAseq experiments, three bronchial brushes were immediately placed in 2 ml of pre-warmed (37 °C) BEGM media. Cells were detached from the brush by flicking the brush. Then, airway epithelial cells were pelleted by centrifugation (250 × $g$, 5 min) and disaggregated by resuspension in 0.05% trypsin/EDTA for 5 min at 37 °C. Trypsinization was stopped by addition of HEPES buffered saline supplemented with 15% FBS, and the cells were again pelleted at 250 × $g$, 5 min. Cells were frozen in freezing

solution (20% DMSO + 80% FBS and BEGM media, 1:1) in liquid nitrogen.

**Sample preparation for single-cell RNA sequencing.** Frozen cell suspensions were thawed at 37 °C, diluted with 20 ml cold (4 °C) PBS (Life Technologies, #14190250) + 10% heat-inactivated FBS (Life Technologies, #10437028), USA), then centrifuged at $300 \times g$ at 4 °C for 5 min. The supernatant was discarded, then the cell pellet was resuspended in PBS + 0.04% BSA (New England Biolabs, # B9000S), and filtered through a 40 µm cell strainer (Fisher Scientific, #50-828-736). For cell concentrations, cells were stained with Trypan blue and counted on the Countess Automated Cell Counter (Thermo Fisher, USA).

**Single-cell barcoding, library preparation, and sequencing.** Single-cell sequencing was performed using on the 10x Genomics Chromium technology according to the manufacturer's protocol (Single Cell 3' Reagent Kits v2, 10x Genomics, USA): Cell suspensions, reverse transcription master mix and partitioning oil were loaded on a single-cell "A" chip, then run on the Chromium Controller. mRNA was reversed transcribed to cDNA within the droplets at 53 °C for 45 min. cDNA was amplified for 12 cycles in total on a BioRad thermocycler. SpriSelect beads (Beckman Coulter, USA) were used for size selection of cDNA based on a ratio of SpriSelect reagent volume to sample volume of 0.6. For quality control, cDNA was analyzed on Agilent Bioanalyzer high-sensitivity DNA chips. cDNA was enzymatically fragmented for 5 min at 32 °C, followed by end repair and A-tailing at 65 °C for 30 min. A double-sided size selection of the cDNA was performed using SpriSelect beads. After adapter ligation, cDNA was amplified using a sample-specific index oligo as primer, followed by a last double-sided size selection using SpriSelect beads. Final cDNA libraries were analyzed again on an Agilent Bioanalyzer high-sensitivity DNA chip. Libraries were sequenced on a HiSeq 4000 Illumina platform aiming for 150 million reads per library.

**Data processing.** Base calls were converted to fastq reads using the wrapper 'mkfastq' of the software Cell Ranger (10x Genomics, USA). Template switch oligo sequence (AAGCAGTGGTATCAACGCAGAGTA-CATGGG) and poly(A) sequence contaminations on Read2 were trimmed with cutadapt (v2.3)[60]. If Read2 sequences were shorter than 30 bp after trimming, read pairs were discarded. zUMIs pipeline (v2.0)[61] was used for subsequent processing of reads. Paired reads were discarded if either the unique molecular identifier (UMI) sequence or the cellular identifier sequence had more than 1 bp with a phred quality score of below 20. Read2 sequences were mapped to the human genome reference GRCh38 release 91 from ensemble[62] using STAR (v2.6.0c)[63]. Reads were collapsed on a UMI level, reads that aligned to both exonic and intronic sequences were retained as both separate and combined gene expression assays. To delineated cell barcodes representative of quality cells from barcodes of apoptotic cells or background RNA, the following thresholds had to be passed: having at least 17.5% of transcripts arising from intronic, i.e., unspliced reads, indicative of nascent mRNA; profiling of more than 3000 UMIs per cell barcode; and <20% of their transcriptome being of mitochondrial origin. To improve the interpretability of the gene identities, ensemble gene IDs were translated to HGNC format only if an exact one-to-one translation was available using the R package BioMart[64].

**Computational analysis.** Raw UMI counts were normalized by scaling to 10,000 UMIs per cell, and then natural log transformed using a pseudocount of 1. Louvain cluster analysis[65] was performed to identify cell types using the R package Seurat (version 3)[66]. Multiplet cell populations, identified as having a transcriptomic gene expression profile that resembled the resulting combination of 2 disparate cell type signatures that already existed in the dataset, and low UMI cell

populations were not included in downstream analyses[14]. Highly variable genes in the dataset were identified as the top 1000 genes ranked by dispersion (scaled variance/mean) across 20 bins of the expression distribution. The expression values for these genes were then adjusted for differences in total UMI and the fraction of mitochondrial reads across cells during z-normalization with a maximum absolute z-score of 10. For linear dimension reduction, these scaled gene expression values were subject to principle component analysis (PCA). A shared nearest neighbor network was created based on Euclidean distances between cells in multidimensional PC space and a fixed number of neighbors per cell, which was used to generate a 2-dimensional Uniform Manifold Approximation and Projection UMAP[67] for visualization. UMAP embeddings of the full dataset are depicted in Supplementary Fig. 4. The dataset was then subsetted to all epithelial populations, re-embedded in UMAP space and colored by cell type and disease status (Fig. 3a, b). To account for subject variability, unity normalized gene expression of canonical marker genes of the epithelial cell types, averaged per subject, grouped by cell type, was visualized in a heatmap (Fig. 3c). Gene expression between groups of cells was compared using a Wilcoxon Rank Sum test with Bonferroni correction of $p$-values for multiple testing. Significantly differentially expressed, upregulated genes in IPF basal cells compared to control basal cells were subjected to a pathway analysis using the software enrichr and the human KEGG 2019 pathway database[68,69].

**Connectivity Map (CMap) analysis to identify pertubagen classes reversing IPF-associated gene expression profiles of ABCs.** Broad Institute's CLUE platform (https://clue.io) was used to identify a potential molecular mechanism of action which could reverse (A) the deviating gene expression profile of ABCs in IPF as identified by scRNAseq in this study and (B) the ABC gene expression signature associated with mortality in IPF by bulk RNA sequencing as recently described by us[13]. CLUE is based on the Connectivity Map dataset[70] that analyses cellular gene expression responses to multiple perturbations. As input, we used (A) the top 150 (the maximum number of inputs in CLUE) genes differentially higher expressed in IPF vs nonUIP ILD Controls in the ABC population of our scSeq dataset ordered by FDR and (B) the top 150 genes associated with mortality from Prasse et al.[13] ordered by FDR. Furthermore, all input genes had to be comprised in the gene space of CLUE (https://clue.io/command?q=/gene-space%20lm). The analysis was run in CLUE's data version 1.1.1.2 and the software version 1.1.1.41. The summary connectivity score, ranging from +100 (pertubagen classes with transcriptional effects similar to the input gene signature) to −100 (perturbagen classes with the opposite effect to the input gene signature, i.e., potential treatments), compares the calculated enrichment score to all others in the reference database of CLUE. Further details are given on https://clue.io/connectopedia/cmap_algorithms and the original publication[70]. The summary connectivity score of all pertubagen classes was extracted, ordered by the summary connectivity score, and visualized as bar plots of the summary connectivity scores split by the origin of the input gene profiles.

**Experimental protocol for the humanized mouse model for IPF.** All animal models were performed multiple times as indicated and data were jointly analyzed. All animals treated were included in the analysis. Interventions were not blinded, but analysis of animal samples was blinded, and, in each experiment, respective controls were included. $Rag2^{-/-}$, NRG and NSG mice received a dose of 1.2 mg/kg bleomycin intratracheally at day 0 during light halothane-induced anesthesia. Three days later $0.3 \times 10^5$ (NRG and NSG $0.2 \times 10^5$) human ABCs from patients with IPF, nonUIP ILD, or healthy volunteers or A549 cells (Merck, #86012804-1VL) were intratracheally injected during light halothane-induced anesthesia. In some experiments lentiviral transduced human ABCs were used (see below). In addition, in some

experiments mice were treated oropharyngeally with 10 mg/kg of saracatinib (kindly provided by Leslie Cousens, AstraZeneca) daily versus vehicle (200 µl PBS) daily. For these experiments, we used pairs of *NRG* mice in which the same human IPF-ABC line was injected. One mouse of the pair was treated with saracatinib and the other with vehicle (PBS). Unless stated otherwise, lungs were harvested at day 21. For histological and immunohistological analyses, the trachea was cannulated, and lungs were insufflated with 4% paraformaldehyde in PBS at a pressure of 25 cm $H_2O$, followed by removal of the heart. Inflated lungs were incubated in 4% formaldehyde solution overnight at 4 °C and after this step 4% formaldehyde solution was replaced by 70% ethanol. Lungs were stored in 70% ethanol at RT until paraffin embedding. For hydroxyproline measurements left and right lungs were weight and homogenized in distilled water on a ULTRA-TURRAX® (VWR, #IKAA3725001) and stored at −80 °C until used.

**Hydroxyproline and total protein assay.** Murine lung hydroxyproline was determined by the hydroxyproline colorimetric assay kit from Biocat (hydroxyproline Kit; Biocat GmbH, #K55-100) following manufacturer's instruction, as previously described[71]. Lung homogenates (100 µl) were mixed with 100 µl 12 N HCl, and the samples were incubated (hydrolyzed) at 120 °C in a pressure-tight, teflon-capped vial (Merck, #Z115096) for 3 h. Afterward, 100 µl of the chloramine T reagent were added to each sample and incubated at RT for 5 min. Then 100 µl of DMAB reagent were added to each well and incubated for 90 min at 60 °C. OD of each sample was determined using the Infinite 200 PRO Tecan microplate reader at a wavelength of 560 nm. Total protein levels were analyzed by the total protein assay kit of Quickzyme (Netherlands, #QZBTOTPROT1) according to the manufacturer's description. Briefly: hydrolyzed lung homogenate samples were diluted 1:2 in 6 M HCL. Then 15 µl of the standard solution (3000 µg/ml high standard, 0.047–3.00 mg/ml) and 15 µl of the samples were added in a 96-well microplate and 120 µl of the assay buffer were added, then mixed on a shaker at 150 rpm for 10 min at RT. Afterward, 15 µl of the color working reagent solution were added to each well. The microplate was covered with an adhesive film and incubated at 85 °C for 60 min in an incubator. OD was determined at 570 nm using the Infinite 200 PRO Tecan microplate reader. Hydroxyproline data are expressed as hydroxyproline [µg/ml]/total protein [mg/ml].

**Hematoxylin-eosin (H&E) staining.** H&E staining was performed according to a standard protocol and as recently described[72]. Sections were incubated in Mayer's hemalum solution (Sigma-Aldrich, #109249) for 2 min followed by rinse in PBS for 3 times and then stained with Eosin Y (yellowish) (C.I. 45380) (Sigma-Aldrich, #115935) in combination with Phloxin B (C.I. 45410) (Sigma-Aldrich, #115926) for 20 s. H&E stainings were carried out in the H&E slide stainer Leica Biosystems #STS5020-005030. Slides were digitalized using Mirax Scan 150 BF/FL (Zeiss, Germany).

**Masson's trichrome staining and fibrosis scoring.** Masson trichrome stains were performed from formalin-fixed paraffin-embedded murine lung tissues according to a standard protocol as recently described[73]. In brief, deparaffinization of paraffin sections mounted on slides was performed in a cuvette with a descending alcohol row: 3 times 100% xylene, 2 times 100% ethanol, once 90% ethanol, once 70% ethanol for 3 min and then incubated in distilled water for 10 min. Then slides were incubated in bouin's solution (150 ml picric acid solution 1.3% (Sigma-Aldrich, #P6744), 50 ml formaldehyde 37% (Carl Roth, #CP10.2), and 10 ml ice acetic acid (Merck, #1.0056) for 15 min at 56 °C. Afterward, slides were cooled at RT and rinsed under tap water. Then slides were incubated in Weigert's solution A and B (Weigert's iron hematoxylin kit; Merck, #1.15973.1) for 10 min. Slides were rinsed under tap water and distilled water and dyed for 5 min in Biebrich scarlet-acid fuchsin

solution (Sigma-Aldrich, #HT151). After another washing step slides were incubated for 10 min in molybdatophosphoric acid solution (Sigma-Aldrich, #1.00532). Afterward, slides were directly transferred to aniline blue Chroma (Sigma-Aldrich, #B8563). Then, slides were rinsed with distilled water, acetic acid, and dehydrated with a descending alcohol row: 70% ethanol 2 min, 96% ethanol 2 min, 100% ethanol 2 min, and 100% xylene 10 min. At the end, slides were mounted with Roti®-Histokitt II (Carl Roth, #160.1). Left lung sections were imaged using Mirax Scan 150 BF/FL (Zeiss, Germany). Ashcroft score was calculated as previously described by Ashcroft and us[74,75]. The scoring scale was as follows: 0 = no abnormalities, 1 = slight thickening of alveolar membranes, 2 = small areas of fibrosis (<10%), 3 = 10–20% fibrotic area, 4 = 20–40% fibrotic area, 5 = 40–60% fibrosis, 6 = 60–80% fibrosis, 7 = >80% fibrosis, and 8 = complete fibrosis.

**Testing engraftment of human ABCs in mouse lungs by bioluminescence imaging.** For in vivo imaging of FLuc transduced human ABCs in NRG mice, 150 mg/kg XenoLight D-Luciferin-K + Salt Bioluminescent Substrate (PerkinElmer, #122799) was injected subcutaneously on day 7, 14, and 21. Mice were anesthetized (1.5% to 2.5% isoflurane) and bioluminescence was measured by an IVIS Lumina II (PerkinElmer, USA). Data were analyzed using LivingImage 4.5 (PerkinElmer, USA).

**Lentiviral vector for firefly-luciferase and enhanced green fluorescent protein (eGFP) overexpression.** The lentiviral vector pRRL.PPT.CBX3-SFFV.FLuc.T2A.eGFP.P2A.Neo.pre (GFP vector) was generated by cloning of the cDNAs for firefly-luciferase (FLuc), enhanced green fluorescent protein (eGFP) and neomycin (Neo) into a lentiviral vector backbone (kindly provided by Dr. Axel Schambach).

**Lentiviral vector for c-src overexpression and knockdown.** The vector pRRL.PPT.CBX3-SFFV.C-SRC.E2AFLuc.T2A.eGFP.P2A.Neo.pre (c-SRC OE vector) for overexpression of c-src was cloned by insertion of a codon-optimized c-src cDNA in front of FLuc into the pRRL.PPT.CBX3-SFFV.FLuc.T2A.eGFP.P2A.Neo.pre vector.

Knockdown of c-src was achieved using an all-in-one CRISPR-Cas9 vector pRRL.PPT.hU6.C-SRC-sgRNA.SFFV.spCas9-2xNLS.T2A.dTomato.pre (c-SRC KO vector) (kindly provided by Dr. Axel Schambach). The sgRNA target sequence for c-src was cloned after phosphorylation and annealing of the oligonucleotides 5′-CACCGAGCGCCGTGCACGTT CTCGG and 5′-AAACCCGAGAACGTGCACGGCGCTC via two BsmBI sites into the all-in-one CRISPR-Cas9 vector.

**Virus production and titration.** Lentiviral vector supernatants were produced by transfection of 10 µg lentiviral vector, 12 µg gag-pol, 6 µg rev, and 2 µg VSVg packaging plasmids into 293T cells (ATCC, LGC Standards, Wesel, Germany) in 10 cm plates. All plasmids were mixed and transfected using calcium-phosphate in the presence of 25 µM chloroquine (Sigma-Aldrich, Seelze, Germany, #C6628). Media exchange was performed 6 h and viral supernatants were harvested 30 and 54 h after transfection. Pooled viral supernatants were filtered through 0.22-µm filters (Millex-GP, Millipore, #SLGP033RS) and 100-fold concentrated using an ultracentrifugation step for 2 h at 25,000 rpm and 4 °C. The infectious titer was determined by applying serial dilutions of viral supernatants onto $10^5$ HT1080 cells (DSMZ, Braunschweig, Germany) in the presence of 4 µg/ml protamine sulfate (Sigma-Aldrich, #P3369).

**Transductions.** For transduction of ABCs and fibroblasts, cells were plated with 30% confluence in T25 flasks 14 days before transduction. ABCs were transduced with c-SRC OE, c-SRC KO, eGFP lentiviral vector supernatants and in addition fibroblasts with eGFP lentiviral vector supernatants. ABCs were cultivated in BEGM media and fibroblasts in DMEM media. Transduction was performed in the presence of 4 µg/ml

protamine sulfate (Sigma-Aldrich, #P3369) using a multiplicity of infection (MOI) of 0.25–0.5. Media exchange was performed 6–8 h after transduction with BEGM and DMEM media as previously described by us[71]. Two days later, transduced cells were selected by applying 1 mg/ml G418 (geneticin, selective antibiotic, Lonza, #15-394N) in BEGM media and DMEM media. Selection media were changed every three days. Luciferase activity of cells was tested using a luminometer prior to use.

**Histology and immunohistochemistry.** Immunohistochemistry (IHC) of formalin-fixed lung tissues from 10 patients with IPF (explants), and 3 healthy lung donors (transplants) was performed. In addition, formalin-fixed mouse lung tissues of the described humanized IPF mouse model and cryopreserved 3D organoids were used for immunohistochemistry. Tissue sections (3 μm) of formalin-fixed human and murine lung tissues embedded in paraffin blocks were deparaffinized in xylene and rehydrated with a descending alcohol row: 3 times 100% Xylene (Merck, #108633), 2 times 100% ethanol (Merck, #107017), once 90% ethanol, once 70% ethanol and distilled water each for 3 min. Heat-induced antigen retrieval was performed at 110 °C for 2 min in citrate buffer (pH 6.0) using a pressure cooker. Prior to staining, slides were rinsed for 5 min with Tris-NaCl buffer and were blocked for 20 min with normal goat serum 1:20 (Vector, #S1000). Incubations with primary antibody were conducted 30 min at RT. All primary antibodies and dilutions are listed below. Afterward, slides were rinsed with Tris-NaCl buffer. Secondary biotinylated antibodies were incubated for 30 min at RT. Afterward, slides were rinsed with Tris NaCl buffer. Activation was done using alkaline phosphatase Strept AP (1:800 dilution) (Vector, #SA-5100) for 30 min and slides were rinsed with Tris-NaCl buffer. Visualization was performed by DAKO REAL Chromogen Red (Dako Real Kit) (Dako/Agilent Technologies, #K500311-2) (incubation time: 20 min). Afterward, slides were rinsed with distilled water. Counterstain with Mayer's hemalum solution (Merck, #109249) 1:10 for 90 s. Slides were rinsed shortly with distilled water and for 90 s with Shandon™ Bluing Reagent (ThermoScientific, #6769001). Before mounting, slides were rinsed once with distilled water followed by 3 times with 90% ethanol, 6 times with 100% ethanol and 6 times with xylene. Finally, slides were coverslipped with Eukitt Quick-hardening mounting medium (Merck, #03989). 3D Organoids were stained directly with three primary antibodies against CK5/6, p63, and αvβ6 integrin. The IHC staining takes place immediately as described above with the exception that the endogenous peroxidase was deactivated by $H_2O_2$ (Merck, #107209) for the second and third primary antibody. Visualization was performed with three different Chromogens DAKO REAL Chromogen Red, DAKO FLEX HRP turquoise Chromogen and EnVision™ FLEX DAB + brown Chromogen DAKO with Peroxidase coupled secondary antibody (DAKO EnVision FLEX/HRP). All slides were digitalized using Mirax Scan 150 BF/FL (Zeiss, Germany). All primary and secondary antibodies that were used for immunohistochemistry are listed in Supplementary Tables 3 and 4.

**Confocal laser scanning microscopy (CLSM).** CLSM of representative cryopreserved 3D organoids w/wo IPF fibroblasts, representative human lung tissue sections and representative murine lung tissue sections was performed. 3-μm-thick sections of paraffin-embedded blocks were deparaffinized with a descending alcohol row as described. Heat-induced antigen retrieval was performed at 110 °C for 2 min in citrate buffer (pH 6.0) using a pressure cooker. At first, staining slides were washed for 5 min with Tris-NaCl buffer and then blocked for 20 min with 4% normal donkey serum (Dianova, #017-000-121). Of cryopreserved 3D organoid cultures 20-μm-thick sections were obtained by Leica CM 1900 (Leica Biosystems, Germany) and directly fixed with acetone (Carl Roth, #5025.1) for 15 min at −20 °C and then dried at RT. Slides were stored at −20 °C until used. Prior to immunofluorescence staining, slides were rehydrated for 5 min with PBS and

then blocked for 15 min with 4% normal donkey serum (Dianova, #017-000-121).

For immunofluorescence analysis, sections were stained with primary and secondary antibodies as listed below. The primary antibodies were incubated at 4 °C overnight in a humidified chamber. On the following day, slides were washed 3 times with 2 ml PBS and the secondary antibodies were incubated for 2 h at RT. Nuclei were stained by DAPI (Sigma-Aldrich, #D9542) as recommended by the manufacturer. Finally slides were washed 3 times with 2 ml PBS and once with 2 ml distilled water and mounted with prolong diamond antifade (Thermo Fisher Scientific, #P36970). Pictures were taken at a laser scanning microscope Olympus FluoView 1000 and with a LSM 510 Meta/Zeiss laser scanning microscope (CLSM). Images were analyzed and processed with software Imaris 7.6 Bitplane Scientific (Oxford Instruments company/USA). All antibodies that were used for CSLM are listed in Supplementary Table 5.

**Protein extraction and western blot analysis.** Proteins from tissues or cultured cells were extracted/lysed in ice-cold RIPA buffer containing 1 M NaCl (Merck, #106404), 1% Nonidet P40 (Merck, #11754599001), 0.5% sodium deoxycholate (Merck, #D6750), 0.1% SDS (Merck, #L3771), 50 mM Tris (Carl Roth, #4855.2) pH 7.4 in ddH$_2$O with protease/phosphatase inhibitor (Cell Signaling, #5872) and incubated on ice for 20 min. Protein concentrations were determined by BCA Kit (Thermo Fisher Scientific, #A53225) as recommended by the manufacturer. Samples were incubated 5 min in Laemmli-sample buffer (BioRad, #1610747) containing 10% β-mercaptoethanol (Merck, #M6250) at 95 °C. Samples were separated using 12% SDS-PAGE gels (BioRad, #671044) and proteins transferred to polyvinylidene difluoride membranes by Trans-Blot® Turbo™ transfer system (BioRad/Germany). After blocking with 5% non-fat dry milk in TBS-T buffer (25 mM Tris-HCl, 150 mM NaCl, 0.1% Tween 20 (Merck, #P1379), pH 7.5), the membrane was incubated overnight at 4 °C with one of the following primary antibodies listed in the table below. GAPDH and α-Tubulin were used for all blots as endogenous control. All primary antibodies were detected using secondary antibody goat anti-rabbit (H + L)-HRP conjugate (1:3000) (BioRad, #1706515) and secondary goat anti-mouse (H + L)-HRP conjugate (1:3000) (BioRad, #1706516). Enhanced chemiluminescence on immunoblot gels (ClarityTM Western ELC Substrate (BioRad, #1705060) was used for detection with ChemiDoc TM MD Imaging System (BioRad/Germany). P-EGF, tot-EGF, fibronectin, α-SMA, collagen type I, P-p44/42, and tot-p44/42 was detected in lysates of healthy donor fibroblasts (Fib). Fib were seeded into a 12-well plate, grown until 80–90% confluence, incubated with serum-free medium overnight, and then exposed to 14 days old conditioned medium of low and big 3D organoids for either 20 min or 48 h and as positive control, for cFn and Type I Collagen, fibroblasts were treated with 10 ng/ml TGF-beta (R&D Systems, #7754-BH-005/CF) for 48 h. Afterward, the cells were directly lysed in 5 × Laemmli buffer containing 10% β-mercaptoethanol and subjected to western blotting. Western Blot was performed as described above with the only exception that a 7.5% and 10% SDS polyacrylamide gels were used. All primary and secondary antibodies that were used for Western blot analyses are listed in Supplementary Tables 6 and 7.

**Statistical analysis**
For not normally distributed data, Mann–Whitney was used for unpaired and Wilcoxon signed rank for paired comparison. When multiple groups were compared, we tested first for normality and then used ANOVA with Tukey correction for multiple testing for unpaired and repeated measures ANOVA for paired data. The exact test used can be found in the respective figure legend. Statistical analysis of the single-cell dataset incl. the software used is described in the methods section, as it is part of the workflow. Throughout, an (adjusted) *p*-value <0.05 was considered significant. Statistical analysis was performed

with GraphPad Prism 9.3.1471 (GraphPad Software Inc). Comparison between groups was done as indicated.

## Reporting summary

Further information on research design is available in the Nature Research Reporting Summary linked to this article.

## Data availability

Sequencing data is available on the GEO repository under the accession number GSE141939. Gene expression of ABCs of IPF and nonUIP ILD controls were compared using a Wilcoxon Rank Sum test with Bonferroni correction of *p*-values for multiple testing. Source data are provided with this paper.

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

## Acknowledgements

The authors thank Stefanie Reuss, Victoria Wirtz, and Angelina Malassa for their technical support. We also thank the following funding sources for their financial support: DZL BREATH, KFO311, Fraunhofer Attract, and a research grant from Astra Zeneca to A.P.; the German Research Foundation (SCHU 3147/1) to J.C.S.; the NIH grants R01HL127349, R01HL141852, U01HL145567, and U01HL122626 to N.K.

## Author contributions

Conceptualization: A.P., B.J., J.C.S., and N.K. Methodology: A.P., B.J., J.C.S., N.A., I.N., M.W., A.S., R.Z., and N.K. Investigation: J.C.S., B.J., N.A., L.P., O.T., T.S.A., G.K., D.J., A.P., D.K., M.W., S.L., W.G., and P.E. Formal analysis: A.P., J.C.S., N.K., B.J., G.K., M.W., A.S., D.J., I.N., and S.L.

Statistical analysis: J.C.S., T.S.A., A.P., B.J., and M.W. Writing—original draft: A.P., J.C.S., and N.K. Writing—review & editing: J.C.S., B.J., N.A., L.P., O.T., T.S.A., G.K., D.J., A.S., D.K., M.W., I.N., H.K., S.L., W.G., P.E., R.Z., N.K., and A.P. Resources: A.P., G.K., D.J., J.C.S., N.K., A.S., W.G., S.L., R.Z., and I.N. Supervision: A.P., J.C.S., and N.K. Funding acquisition: A.P., N.K., and J.C.S.

## Funding

## Competing interests

The authors declare the following competing interests: N.K. is or has been a consultant to Biogen Idec, Boehringer Ingelheim, Third Rock, Pliant, Samumed, NuMedii, Indalo, Theravance, LifeMax, Three Lake Partners, Optikira, and received non-financial support from Miragen and NuMedii. In addition, N.K. has patents on New Therapies in Pulmonary Fibrosis, Cell Targeting and Peripheral Blood Gene Expression N.K. and J.C.S. are inventors on a provisional patent application (62/849,644) submitted by NuMedii, Inc., Yale University and Brigham and Women's Hospital, Inc. that covers methods related to IPF-associated cell subsets. A.P. is a consultant to Boehringer Ingelheim, Roche, Pliant, Indalo, Nitto Denko and Astra Zeneca. A.P. and J.C.S. have patents on New Therapies in Pulmonary Fibrosis. The following authors declare no competing interests: B.J., L.P., O.T., N.A., G.K., P.E., T.S.A., R.Z., H.K., S.L., W.G., I.N., D.J., M.W., D.K., and A.S.

## Additional information

[1]Fraunhofer Institute for Toxicology and Experimental Medicine, Hannover, Germany. [2]German Center for Lung Research, BREATH, Hannover, Germany. [3]Section of Pulmonary, Critical Care and Sleep Medicine, Yale School of Medicine, New Haven, CT, USA. [4]Department of Pulmonology, Hannover Medical School, Hannover, Germany. [5]Institute of Surgical Pathology, University Medical Center, Freiburg, Germany. [6]Department of Pneumology, University Medical Center, Freiburg, Germany. [7]Leibniz Research Laboratories for Biotechnology and Artificial Organs, Hannover Medical School, Hannover, Germany. [8]Institute for Laboratory Animal Science, Hannover Medical School, Hannover, Germany. [9]Institute for Infection Prevention and Hospital Epidemiology, Medical Center - University of Freiburg, Freiburg, Germany. [10]German Cancer Consortium (DKTK), Partner Site Freiburg and German Cancer Research Center (DKFZ), Heidelberg, Germany. [11]Institute of Pathology, Hannover Medical School, Hannover, Germany. [12]Department of Biochemistry, Faculty of Medicine, Justus Liebig University, Gießen, Germany. [13]Institute of Experimental Hematology, Hannover Medical School, Hannover, Germany. [14]Division of Hematology/Oncology, Boston Children's Hospital, Harvard Medical School, Boston, MA, USA. [15]These authors contributed equally: Benedikt Jaeger, Jonas Christian Schupp. ✉e-mail: prasse.antje@mh-hannover.de

