## [Peer Review File · Nature Communications]

Reviewers' Comments:

Reviewer #1:

Remarks to the Author:

Re: NCOMMS-20-40313-T

In this manuscript, the authors explore distinct properties of airway basal cells isolated from patients with idiopathic pulmonary fibrosis (IPF). This manuscript follows up the group's work studying IPF at a single cell resolution where they identified an aberrant basaloid KRT17+ population enriched in patients with IPF. The authors combined single cell RNAseq, organoid and mouse models to identify the functional role of these human IPF-ABCs, which has previously not been shown before. Further analysis of the single cell data revealed enrichment of src signalling which when inhibited resulted in blocked bronchialisation in vitro and in vivo. This manuscript describes a novel mouse model to study IPF and the further validation of ABCs for therapeutic intervention in IPF.

Critique

The study clearly shows a functional role of human IPF-ABCs in vitro and in vivo, and the experiments are well performed. There are many loose ends making the manuscript in its present form not suitable for publication in Nature Communications.

Major comments:

1. Do these IPF-ABCs in the scRNAseq express similar markers to the published aberrant basaloid cells from the same group, Adams et al Sci Adv 2020? On a related note, the previous papers show a clear separation of the basaloid cells, can the authors comment as to why they do not observe this here?
2. Can the authors comment on the control patient with a gene signature indicative of IPF-ABCs? The previous paper noted that no control patient had any evidence of aberrant basaloids.
3. Can the authors demonstrate where the key genes that mark the IPF-ABCs are expressed in the UMAP?
4. Can the authors comment as to the characteristics of IPF-ABCs in vitro? Do they still express key markers found in vivo?
5. Although the authors mention that they find bronchospheres from all three conditions, Fig. 2c and d indicate that HV ABCs did not form any spheres. How can the authors explain this? Isolated airway epithelial cells from HV usually result in spheres in a 3D-matrigel assay.
6. In Fig 2.f, the authors measure Amphiregulin levels at D14, can the authors comment as to why they chose this time point and not D21?
9. Is the level of Amphiregulin a measure of sphere number, ie: the more spheres, the more Amphiregulin? Can the authors comment on this?
10. Are there any functional differences in addition to counts and proliferation observed when using IPF-ABCs compared to HV-ABCs in 3D organoids culture? What about differentiation markers?
11. In the legend of Fig 2, the authors mention an increase in col1 and SMA in high BS compared to low BS. The figure does not reflect this statement and is not mentioned in the text. How many replicates were performed for this experiment?
12. The authors use CRISPR to knockout, however text and figures swap between knockout and knockdown, can the authors stay consistent? Also, Fig.4f depicts a knockout, but with WT not expressing very little detectable protein, how can the authors state that there is knockout? Can they confirm the presence of an indel?

13. Does the src KO or Ovx have an effect on the bronchosphere assay?

Minor points

1. The authors need to stay consistent with naming these IPF-ABCs as they also use basal cells in the text.
2. How long are the IPF-ABCs kept in culture before moving to 3D - matrigel assay?
3. No scale bar in figure 2b in the lower two panels
4. Fig. 2f - The r value in this graph is inconsistent with the text.
5. In figure 2m/n, authors mentioned they stimulate lung fibroblasts, which ones? HV or IPF?
6. In figure 2f legend: h,k,i instead of h,j,k
7. The authors state that similar data (to bronchosphere counts) was observed in the MTT cell proliferation assay after saracatinib, however 75nm had a bigger impact on counts than cell proliferation. Can the authors change text to reflect this?

Reviewer #2:

Remarks to the Author:

This manuscript by Benedikt Jaeger et al. described that "Airway Basal Cells show a dedifferentiated KRT17^{high} Phenotype and promote Fibrosis in IPF". The authors demonstrated that the levels of c-Src expression was higher in the patients of IPF leading to the expression of fibrotic markers. C-Src seemed to play a key role in these responses. The authors tried to identify the role by either overexpression or downregulation of c-Src compared to WT cells. On the basis of results presented above, the levels of c-Src should be higher than that of normal ABCs. In Fig 4f, I wondered why the levels of c-Src seemed to disappear on WT, similar to the downregulation by KO treatment. This problem may cause a critical concerning about the role of c-Src in these responses. I thought some more experiments required for clarification of the question.

Reviewer #3:

Remarks to the Author:

This manuscript, entitled "Airway Basal Cells show a dedifferentiated KRT17^{high} Phenotype and promote Fibrosis in IPF", describes the characterization of ABCs (airway basal cells) in IPF (idiopathic pulmonary fibrosis) patients. They further focused on the Src gene as a possible therapeutic target of IPF and demonstrated its usage by using in vitro and mouse models. The paper roughly consists of three parts. In the first part, the authors conducted scRNA seq analysis. They demonstrated that ABCs of IPF patients (IPF-ABCs) show distinct molecular profiles from those of nonUIP ILD controls. Particularly, IPF-ABCs showed an extensive re-programming towards KRT17^{high} and PTEN^{low} cells. In the second part, the authors employed the 3D organoid model. IPF-ABCs formed characteristic structure reflecting bronchosphere formation. This was not observed from ABCs of healthy volunteers (HV-ABCs) or nonUIP ILD patients (nonUIP ILD-ABCs). They also found that IPF-ABCs were able to induce proliferation to co-cultured fibroblasts. In the in vivo mouse model, Intra-tracheal application of IPF-ABCs, but not control ABCs, caused sever fibrosis to the bleomycin-injured model mice. For the third part, the authors focused on the Src expression, which was identified from their in silico analysis of Connectivity Map dataset. They found that a SRC inhibitor, saracatinib, demonstrate a favorable effect to the profibrotic changes in the 3D culture and mice models. Overall, I consider this manuscript should contain substantial useful information towards developing a new therapeutic mean for IPF. However, I'm not sure how the Src should be rationalized as a drug target from the results of the scRNA seq analysis.

Major point:

1 At least to me, it is not clear how the knowledge obtained from the scRNA seq analysis and 3D culture assays should provide rationale for the Src inhibition.

2 Possible mode of action of the saracatinib should be examined in a more concrete manner. Otherwise, it is not clear how the former part of the paper should be associated with the latter.

3 How the Src inhibition should change the gene expression program in IPF-ABCs should be further characterized in vitro and in vivo. The effects on the fibroblast side should also be evaluated.

3 How the in silico analysis of Connectivity Map dataset is conducted should be described in more details. The database can be also used to see how the Src inhibitor in contrast to other inhibitors show its activity in other cell types.

4 The results of the scRNA analysis are interpreted only superficially. Further attempts should be made to clarify how the observed gene expression pattern is realized by what molecular signals.

Minor point:

5 For the scRNA, the procedure of the data analysis should be described in more details. The normalization/quality control and the batch-effect removal methods are almost always an issue of discussion.

6 Was not it possible to conduct a validation analysis of DGEs identified from scRNA in the latter experiments in vitro? Conversely, were the molecular features represented in Fig2m and 2n represented in the scRNA analysis?

7 Discussion section is somewhat redundant and could be shortened.

8 Instead, the discussion would be enriched by including additional information. Possible involvements of Src (or Src family kinases) have been argued in a number of papers. Please include further references for this issue. Also I noticed that a clinical trial of the use of saracatinib for IPF is on-going, in which the authors themselves also seem to participate. Further provisions, as far as possible, towards the clinical application of this drug would be very much appreciated.

9 It would be interesting to conduct a spatial transcriptome analysis for the mouse model used in Fig3. The local interaction between ABCs and Fibroblasts may be revealed by such an analysis.

Reviewer #4:

Remarks to the Author:

This is an important study showing the unique changes in airway basal cell phenotype in patients with IPF and the in vitro and in vivo influence of these cells on the fibrotic response, using a variety of experimental models. I think that there are some major methodological issues that should be fixed in order to make the paper more convincing.

Major comments :

1- The authors used purified ABC in the organoid model in vitro and in the bleomycin model in vivo. The authors should describe the methods used for purification and the purity of the cell population that was used. I suspect that there is huge heterogeneity in the cells obtained and injected, which really impacts the understanding of your experiments and your results.

2-In these in vitro and in vivo experiments, an important control condition is missing, with epithelial cells of a different origin (alveolar epithelial cells, or ciliated epithelial cells) in order to demonstrate that the effect of ABC is specific for ABC, or is just epithelial-cell dependent.

3-The control group should include a UIP-fibrotic lung disorder to compare with idiopathic-UIP/IPF, at least in some in vivo fundamental experiments. Indeed, the identification of a specific phenotype in IPF as compared with non UIP fibrotic ILD raise the important question of a UIP-

specific or an IPF-specific ABC phenotype.

4- The identification of Src inhibition as a potential therapeutic target in lung fibrosis is not new (Hu, JPET 2014) and it is not surprising to see that Saracatinib inhibits lung fibrosis (Figure 6). The original result here is that Src activation might be specific to the ABC. However, the data provided by the authors to support that conclusion are limited. Indeed, immunofluorescence data in the fibrotic lung (Figure 4c and Figure S5) show only a few red dots on a faint red background (Fig S5). Interestingly, these data contrast with a strong and diffuse immunolabeling of the epithelial cells with immunohistochemistry. The discrepancy between methods is intriguing.

5- Saracatinib is being used in the current manuscript as a src inhibitor. However, Saracatinib is a SFK/Abl dual-kinase inhibitor. The Abl kinase is expressed in the lung epithelium and may be involved in lung repair after injury (Khatri, PNAS 2019, PMID 30655340). Interpreting the pharmacological results should be cautious.

6-The statistical methods in the methods section is poor. The authors did use in some experiments a t-test where non parametric tests should be used due to the limited number of experiments performed.

Minor comments

1-Figure 4b : please indicate the control and IPF tissue in the figure

2-Figure 4c and Figure S5 : the authors state that c-src is highly expressed in the fibrotic lung. This is difficult to prove on the basis of the immunofluorescence pictures provided: this reviewer sees only a few red dots on a faint red background (Fig S5).

3- Figure 6 : in the in vivo experiments, the effect of saracatinib on bleomycin induced lung fibrosis (without ABC injection) is missing.

4-This reviewer would like to see the effect of src inhibition in human lung slices (control and IPF) as this is a unique full-human model, that would nicely complement the in vitro and in vivo experiments. This is important because the relevance of the organoid model to the pathophysiology of IPF is at best undetermined.

Point by Point Response to Reviewers' comments

REVIEWER COMMENTS

Reviewer #1 (Remarks to the Author):

Re: NCOMMS-20-40313-T

In this manuscript, the authors explore distinct properties of airway basal cells isolated from patients with idiopathic pulmonary fibrosis (IPF). This manuscript follows up the group's work studying IPF at a single cell resolution where they identified an aberrant basaloid KRT17+ population enriched in patients with IPF. The authors combined single cell RNAseq, organoid and mouse models to identify the functional role of these human IPF-ABCs, which has previously not been shown before. Further analysis of the single cell data revealed enrichment of src signalling which when inhibited resulted in blocked bronchialisation in vitro and in vivo. This manuscript describes a novel mouse model to study IPF and the further validation of ABCs for therapeutic intervention in IPF.

Critique

The study clearly shows a functional role of human IPF-ABCs in vitro and in vivo, and the experiments are well performed. There are many loose ends making the manuscript in its present form not suitable for publication in Nature Communications.

Major comments:

1. Do these IPF-ABCs in the scRNAseq express similar markers to the published aberrant basaloid cells from the same group, Adams et al Sci Adv 2020? On a related note, the previous papers show a clear separation of the basaloid cells, can the authors comment as to why they do not observe this here?

R1.1: We thank the reviewer for this question. The IPF-ABCs are not the aberrant basaloid cells described by us in single cell RNAseq of human lung tissues. The IPF-ABCs of this publication do share some features with aberrant basaloid cells like high expression of KRT17; integrin subunits ITGAV, ITGB6, ITGB8; CD24 and ADAM9. However, they do not share other features; they express KRT5, and they lack other markers such as CDKN2A, CDKN2B, PRSS2, CTSE, MDM2, TGBFI, SLCO2A1, and MMP7. In the dataset of this publication analyzing bronchial brushings of IPF patients and nonUIP disease controls we were not able to detect any aberrant basaloid cells.

2. Can the authors comment on the control patient with a gene signature indicative of IPF-ABCs? The previous paper noted that no control patient had any evidence of aberrant basaloids.

R1.2: We thank the reviewer for this answer. Again it is important to highlight that IPF-ABC are not aberrant basaloid cells. The control patients in this dataset are disease controls and not healthy volunteers and the classification of UIP or non UIP was based on HRCT findings. Some fibrotic ILD patients with a pattern on HRCT that is inconsistent with UIP, do exhibit a UIP pattern on histology or develop later in their course of disease a UIP pattern. Therefore, it is conceivable that some disease control patient will exhibit the molecular phenotype that is similar to the IPF/UIP patients in this dataset. Our goal of the scRNAseq study was to generate hypotheses on cell type specific

molecular pathomechanisms and targets. We did not intend to establish a biomarker that distinguishes UIP from non UIP. We provide a better description of the patients in the supplement now.

3. Can the authors demonstrate where the key genes that mark the IPF-ABCs are expressed in the UMAP?

R1.3: We apologize for the missing UMAP visualization of key genes of IPF-ABCs. We have now included the genes in supplemental Fig. S2.

Figure 1: Expression of selected key genes of (IPF-)ABCs genes. UMAPs of epithelial cells colored by gene expression values of features that are associated with basal cells or with increased expression in IPF-ABCs compared to controls.

4. Can the authors comment as to the characteristics of IPF-ABCs in vitro? Do they still express key markers found in vivo?

R1.4: We thank the reviewer for this remark. ABC do change in culture. When we checked for basal cell markers after 21 days of cell culture using BEGM as previously published (Hackett et al. PLoS One 2011, doi: 10.1371/journal.pone.0018378), more than 97% of all cells expressed KRT5 and KRT17 (see Figure 2 below). We added this information to the supplemental methods. These ABCs were then used for our 3D organoid model and the mouse model. Naturally, bronchosphere formation is associated with cell proliferation, migration and differentiation. After 21 days of organoid culture only a subset of the sphere forming cells still expressed markers of ABC such as KRT5 and these cells still express other key markers such as ITGB6 (as demonstrated in Fig. 2c in the manuscript) and KRT17. Of interest, KRT5+ABC are located at the outer rim of the spheres, which is in contact to the extracellular matrix of the matrigel mimicking basement membrane. As shown in Fig. 2b and c of the manuscript the center of the sphere gets hollow over time and the inner layer of spheres is composed out of differentiated airway epithelial cells.

Figure 2: Outgrowth of ABCs were analyzed by flow cytometry after 21 days of cell culture. Depicted are representative original registrations of intracellular stainings and gating strategy. Almost all cells expressed KRT5 and KRT17.

5. Although the authors mention that they find bronchospheres from all three conditions, Fig. 2c and d indicate that HV ABCs did not form any spheres. How can the authors explain this? Isolated airway epithelial cells from HV usually result in spheres in a 3D-matrigel assay.

R1.5: We thank the reviewer for bringing this up. Indeed ABC from most healthy volunteers do not produce any spheres consisting of more than 3 cells and have a diameter less than 50 μ m. However, some ABC lines produced small numbers of small spheres. This is exactly what is shown in the figures – HV-ABC generate a very small amount of relatively small bronchospheres, whereas IPF-ABC resulted in higher amounts and more complex bronchospheres. Isolated HV-ABCs do not usually give rise to bronchospheres applying basic protocols of sphere forming assays. Our protocol is based on previously published publications and uses a transwell cell culture system. Cells are placed in matrigel on top of the transwell and cultured in a commercially available medium developed for the

growth of airway epithelial cells. Several groups (Hegab AE et al. *Stem Cells Dev.* 2014, doi: 10.1089/scd.2013.0295; Rock JR et al. *Dis Model Mech.* 2010, doi: 10.1242/dmm.006031) have used a similar bronchosphere model as ours and found that primary normal murine or human KRT5⁺ ABCs do not generate a substantial amounts of bronchospheres (<3%). Prior enrichment steps for specific subpopulations of basal cells (e.g. ITGB6⁺), presence of feeder cells or adding factors like noggin and EGF (Salahudeen AA et al. *Nature* 2020, doi.org/10.1038/s41586-020-3014-1) increases numbers of bronchospheres. Surprisingly, we observed that KRT5⁺ IPF-ABCs do not require feeder cell support, but they definitely profit from the crosstalk with fibroblasts or feeder cells. Of note IPF-ABCs express higher levels of ITGB6 as we have shown by scRNAseq and may be therefore enriched for a population with higher sphere-forming capacities.

6. In Fig 2.f, the authors measure Amphiregulin levels at D14, can the authors comment as to why they chose this time point and not D21?

R1.6: Conditioned media were harvested every 7 days. Amphiregulin levels were tested at every time point. We noted an increase in amphiregulin levels over time and this correlates with bronchosphere counts as indicated in Fig. 2f of the manuscript. We included the data of d14 because we were interested in factors that may drive early events of sphere formation and do not just reflect cell numbers.

Figure 3: Human ABCs were cultured in the described 3D organoid model. Conditioned media were harvested at day 7 (d7), Day 14 (d14), and d21 (d21). Amphiregulin levels were detected by ELISA.

7. Is the level of Amphiregulin a measure of sphere number, ie: the more spheres, the more Amphiregulin? Can the authors comment on this?

R1.7: We thank the reviewer for these questions. As described in our response to the reviewer's 6th comment, we decided to measure earlier time points to see whether increases in amphiregulin levels preceded the increase in numbers of bronchospheres. Amphiregulin is an EGFR ligand, which induces cell proliferation, and is highly produced by IPF-ABCs. We hypothesize, that amphiregulin levels in IPF-ABC organoid cultures are not only high because of higher cell numbers, but rather drive higher basal cell proliferation in IPF. We were, however, cautious in our statements since the setup of our experiment does not allow to dissect whether amphiregulin levels drive cell proliferation or just reflect cell counts.

8. Are there any functional differences in addition to counts and proliferation observed when using IPF-ABCs compared to HV-ABCs in 3D organoids culture? What about differentiation markers?

R1.8: Our co-culture experiments clearly show that IPF-ABCs in contrast to HV-ABCs promote fibroblast proliferation and collagen production in addition to increased proliferation and sphere counts. Beyond that we have not tested other functional effects of IPF-ABCs yet. Since HV-ABC do not generate fully developed bronchospheres in our model, we were not able to compare subsequent cell differentiation of IPF-ABCs versus HV-ABCs.

9. In the legend of Fig 2, the authors mention an increase in col1 and SMA in high BS compared to lowBS. The figure does not reflect this statement and is not mentioned in the text. How many replicates were performed for this experiment?

R1.9: We thank the reviewer for raising this concern. We repeated the experiment with conditioned media of ABCs with low bronchosphere generation and IPF-ABCs with high bronchosphere generation and added the Western blot of this experiment to Fig. 2m. In total, we tested n=11 different samples and quantified protein expression which was also added to Fig. 2n and o.

10. The authors use CRISPR to knockout, however text and figures swap between knockout and knockdown, can the authors stay consistent? Also, Fig.4f depicts a knockout, but with WT not expressing very little detectable protein, how can the authors state that there is knockout? Can they confirm the presence of an indel?

R1.10: We thank the reviewer for this comment. Indeed the depicted Western blot was confusing. The overexpression worked so well that the band of the WT sample was hardly depicted anymore. We performed an additional Western blot of the same samples (revised Fig. 4f and Fig. S10). The reviewer is right that we should be consistent in our wording. Using lentiviral transduction we obtained a knockdown for src protein expression and not a knockout as indicated in Fig.4. We corrected the wording of our manuscript and the respective figure legends.

11. Does the src KO or Ovx have an effect on the bronchosphere assay?

R1.11: Again, we thank the reviewer for this suggestion that definitely improved our manuscript. We performed the requested experiments and added the results to Fig. 4 as panels g and h. Overexpression of c-src in ABCs increases sphere counts. Knockdown decreased sphere counts compared to WT.

Minor points

12. The authors need to stay consistent with naming these IPF-ABCs as they also use basal cells in the text.

R1.12: As suggested by the reviewer, we changed the wording of the manuscript and stay now consistent in naming the cells ABCs.

13. How long are the IPF-ABCs kept in culture before moving to 3D - matrigel assay?

R1.13: Outgrowth of ABCs from bronchial brushings was achieved after 21 days. Then cells were immediately used for the 3D matrigel assay. ABCs were not kept in culture or passaged prior to the 3D matrigel assay.

14. No scale bar in figure 2b in the lower two panels.

R1.14: We thank the reviewer for this notification. We had incorporated a small-scale bar in both panels, but indeed it was too small. We increased the size of the scale bars to improve readability.

15. Fig. 2f - The r value in this graph is inconsistent with the text.

R1.15: We thank the reviewer for this notification. The text stated r^2 while the text in the figure stated r value. In order to stay consistent we now state only r^2 values at the figure and manuscript.

16. In figure 2m/n, authors mentioned they stimulate lung fibroblasts, which ones? HV or IPF?

R1.16: In this experiment donor (HV) fibroblast were used. The information was added to the legend.

17. In figure 2f legend: h,k,i instead of h,j,k

R1.17: We thank the reviewer for the thorough review of our manuscript and corrected the mistake in the figure legend of Figure 2.

18. The authors state that similar data (to bronchosphere counts) was observed in the MTT cell proliferation assay after saracatinib, however 75nm had a bigger impact on counts than cell proliferation. Can the authors change text to reflect this?

R1.18: We thank the reviewer for this comment and revised our statement in regards to the effect of saracatinib on cell proliferation.

Reviewer #2 (Remarks to the Author):

This manuscript by Benedikt Jaeger et al. described that “Airway Basal Cells show a dedifferentiated KRT17^{high} phenotype and promote Fibrosis in IPF”. The authors demonstrated that the levels of c-Src expression was higher in the patients of IPF leading to the expression of fibrotic markers. C-Src seemed to play a key role in these responses. The authors tried to identify the role by either overexpression or downregulation of c-Src compared to WT cells. On the basis of results presented above, the levels of c-Src should be higher than that of normal ABCs. In Fig 4f, I wondered why the levels of c-Src seemed to disappear on WT, similar to the downregulation by KO treatment. This problem may cause a critical concern about the role of c-Src in these responses. I thought some more experiments required for clarification of the question.

R2: We thank the reviewer for this comment and agree that the presented Western blot was suboptimal. Lentiviral transduction overexpressed indeed c-src. There is a c-src band in WT cells but much lower than in transduced, c-src-overexpressing cells. As already noted in response to reviewer's 1 10th comment we improved the Western blot and added functional experiments which demonstrate that c-src overexpression leads to increased sphere counts, while knock down reduces

sphere counts. We would also like to note that culture conditions and cellular microenvironment have an influence on c-src expression. Extracellular matrix products stimulate c-src expression via multiple pathways e.g. focal adhesion kinase and integrin signaling.

Reviewer #3 (Remarks to the Author):

This manuscript, entitled “Airway Basal Cells show a dedifferentiated KRT17^{high} Phenotype and promote Fibrosis in IPF”, describes the characterization of ABCs (airway basal cells) in IPF (idiopathic pulmonary fibrosis) patients. They further focused on the Src gene as a possible therapeutic target of IPF and demonstrated its usage by using in vitro and mouse models. The paper roughly consists of three parts. In the first part, the authors conducted scRNA seq analysis. They demonstrated that ABCs of IPF patients (IPF-ABCs) show distinct molecular profiles from those of nonUIP ILD controls. Particularly, IPF-ABCs showed an extensive re-programing towards KRT17^{high} and PTEN^{low} cells. In the second part, the authors employed the 3D organoid model. IPF-ABCs formed characteristic structure reflecting bronchospere formation. This was not observed from ABCs of healthy volunteers (HV-ABCs) or nonUIP ILD patients (nonUIP ILD-ABCs). They also found that IPF-ABCs were able to induce proliferation to co-cultured fibroblasts. In the in vivo mouse model, Intra-tracheal application of IPF-ABCs, but not control ABCs, caused sever fibrosis to the bleomycin-injured model mice. For the third part, the authors focused on the Src expression, which was identified from their in silico analysis of Connectivity Map dataset. They found that a SRC inhibitor, saracatinib, demonstrate a favorable effect to the profibrotic changes in the 3D culture and mice models. Overall, I consider this manuscript should contain substantial useful information towards developing a new therapeutic mean for IPF. However, I’m not sure how the Src should be rationalized as a drug target from the results of the scRNA seq analysis.

Major point:

1 At least to me, it is not clear how the knowledge obtained from the scRNA seq analysis and 3D culture assays should provide rationale for the Src inhibition.

R3.1: The rationale to test the effect of c-src inhibition was built on the connectivity map analyses of our scRNAseq dataset and the recently published BAL dataset. Of interest, this finding is also in line with the independent analyses performed by Dr. Kaminski’s and Joel Dudley’s teams based on the transcriptome of lung tissue samples. This manuscript (Ahangari F et al.) is currently under revision at the Am J Respir Crit Care Med and provided in copy.

2 Possible mode of action of the saracatinib should be examined in a more concrete manner. Otherwise, it is not clear how the former part of the paper should be associated with the latter.

R3.2: We thank the reviewer for this excellent suggestion. We performed the requested experiments and tested inhibition of c-src phosphorylation by saracatinib. Indeed, saracatinib reduced src phosphorylation as shown by Western blot. The data were added to the supplement (Fig.S11).

Figure 4: Down-regulation of phosphorylated c-src by saracatinib treatment in IPF-ABCs. Depicted is the cropped blot of three different IPF-ABC lines which were either untreated or saracatinib treated.

3 How the Src inhibition should change the gene expression program in IPF-ABCs should be further characterized in vitro and in vivo. The effects on the fibroblast side should also be evaluated.

R3.3: We have not analyzed the changes in gene expression of IPF-ABCs by saracatinib treatment. We have, however done this for fibroblasts and this dataset is part of a manuscript, which was submitted in parallel at the Am J Respir Crit Care Med. This manuscript, already mentioned in R3.1, also assesses effects on fibroblasts and is attached.

4 How the in silico analysis of Connectivity Map dataset is conducted should be described in more details. The database can be also used to see how the Src inhibitor in contrast to other inhibitors show its activity in other cell types.

R3.4: We thank the reviewer for mentioning this. The methods section in the supplement of the Connectivity Map analysis was expanded to make this clearer. Regarding the activity of src inhibition in other cell types. This is indeed of interest but beyond the scope of this manuscript that highlights the potential profibrotic role of ABCs in lung fibrosis. We are not claiming that other cell types are not important, but highlighting the role of one cell type, and the potential importance of SRC activation in this cell.

5 The results of the scRNA analysis are interpreted only superficially. Further attempts should be made to clarify how the observed gene expression pattern is realized by what molecular signals.

R3.5: We thank the reviewer for this important remark. We agree that scRNAseq data are a valuable resource that can be analyzed from many different perspectives. Our targeted approach of the scRNAseq study was to generate hypotheses on cell type-specific molecular pathomechanisms and potential therapeutic targets. Nonetheless, we do assume that many other research questions can be answered by this data (beyond the scope of this publication) and this is why we deposited the data on GEO (accession number GSE141939) so that other researchers can use this data as well.

As an example of a more sophisticated approach, we indeed observed subpopulations within IPF-ABCs as seen in Figure 5 below. First, as recently described by Carraro et al. (Carraro G et al. Nat Med 2021, doi: 10.1038/s41591-021-01332-7), we observed a KRT15+DST+KRT5+ canonical basal cell population (Basal_1) and a SERPIN-high (SERPINB4, SERPINB3, Basal_3) as well. However, in IPF, we observed additional diversity with ABCs: a KRT6A+,ITGB8-high, ITGAV-high IPF-ABC (Basal_4) population seems to be distinguishable from a KLF4+,MYC+,CTGF+IPF-ABC (Basal_2) population. However, as these two extremes also align with specific patients, it is not clear to us whether these differences “only” represent subject-specific difference or true IPF-ABC archetypes. Our study does not included enough patients to answer this important question, why we chose to omit any speculation about this.

Figure 5: Basal cell subpopulations. A) UMAP colored by cluster membership of basal cell subpopulations and other epithelial cells. B) Violin Plots of selected marker genes of basal cell subpopulations, split and colored by cluster membership of basal cell subpopulations.

Minor point:

5 For the scRNA, the procedure of the data analysis should be described in more details. The normalization/quality control and the batch-effect removal methods are almost always an issue of discussion.

R3.5: The methods section in the supplement of the scRNA data analysis was expanded. Batch-effect removal or integration approaches have not been performed because all samples were processed in one batch. As described in the supplemental methods, raw UMI counts were normalized with a scale factor of 10,000 UMIs per cell, and then natural log transformed using a pseudocount of 1. To ensure that only high-quality cells are analyzed, cell barcodes from apoptotic cells or background RNA were removed if they had less than 17.5% of transcripts arising from intronic, i.e. unspliced reads, indicative of nascent mRNA; less than 3000 UMIs per cell barcode; and more than 20% of their transcriptome being of mitochondrial origin.

6 Was not it possible to conduct a validation analysis of DGEs identified from scRNA in the latter experiments in vitro? Conversely, were the molecular features represented in Fig2m and 2n represented in the scRNA analysis?

R3.6: We thank the reviewer for these suggestions. Fig. 2f and Fig.2g of the manuscript shows increased production of AREG and CCN2 (CTGF) by IPF-ABCs. Both genes are also increased in IPF-ABCs compared to disease control in the scRNAseq dataset. Fig. 2m and 2n of the original version of the manuscript showed protein expression of lung fibroblasts cultured in conditioned media of bronchospheres. Our scRNAseq dataset did not contain fibroblasts.

7 Discussion section is somewhat redundant and could be shortened.

R3.7: We thank the reviewer for mentioning this and revised the discussion section.

8 Instead, the discussion would be enriched by including additional information. Possible involvements of Src (or Src family kinases) have been argued in a number of papers. Please include further references for this issue. Also I noticed that a clinical trial of the use of saracatinib for IPF is on-going, in which the authors themselves also seem to participate. Further provisions, as far as possible, towards the clinical application of this drug would be very much appreciated.

R3.8: We thank the reviewer for these excellent suggestions. We included further references in regards to the involvement of src in IPF. We mention now also the clinical trial testing the effectiveness of saracatinb in IPF.

9 It would be interesting to conduct a spatial transcriptome analysis for the mouse model used in Fig3. The local interaction between ABCs and Fibroblasts may be revealed by such an analysis.

R3.9: We agree with the reviewer that spatial transcriptomics would be of high interest, but think that this is beyond the scope of our current manuscript. In this context, we would also like to mention our recent publication (Prasse A et al. Am J Respir Crit Care Med 2019, doi: 10.1164/rccm.201712-2551OC) which includes immunohistochemistry and demonstrates that KRT5+ABCs cover almost all fibroblast foci and highly accumulate in IPF tissues.

Reviewer #4 (Remarks to the Author):

This is an important study showing the unique changes in airway basal cell phenotype in patients with IPF and the in vitro and in vivo influence of these cells on the fibrotic response, using a variety of experimental models. I think that there are some major methodological issues that should be fixed in order to make the paper more convincing.

Major comments :

1- The authors used purified ABC in the organoid model in vitro and in the bleomycin model in vivo. The authors should describe the methods used for purification and the purity of the cell population

that was used. I suspect that there is huge heterogeneity in the cells obtained and injected, which really impacts the understanding of your experiments and your results.

R4.1: We thank the reviewer for raising these concerns. ABCs were propagated from bronchial brushings using BEGM medium, as published by Hackett et al. (PLoS One 2011, doi: 10.1371/journal.pone.0018378). After 21 days of cell culture in tissue flasks > 97% of all cells expressed KRT5 and KRT17 as shown in Figure 2 and the reply to reviewer's 1 fourth comment. We added this information to the supplemental methods. We agree with the reviewer that heterogeneity within the ABC population would explain why IPF-ABCs are so different from HV-ABCs. Our scRNAseq dataset of bronchial brushings identified substantial changes in gene expression but actually did not confirm a dramatic difference in cell heterogeneity. Thus while we cannot completely rule it out, it seems that the majority of changes in IPF-ABC were not related to population diversity and emergence of novel populations, but instead to global phenotypic changes.

2-In these in vitro and in vivo experiments, an important control condition is missing, with epithelial cells of a different origin (alveolar epithelial cells, or ciliated epithelial cells) in order to demonstrate that the effect of ABC is specific for ABC, or is just epithelial-cell dependent.

R4.2: We thank the reviewer for this suggestion. To control for the effect of epithelial cells per se we used A549 cells. In line with our recently published data (Jäger B et al. Cell Signal 2020, doi: 10.1016/j.cellsig.2020.109672) injection of A549 into our model did not promote any fibrosis. In contrast to IPF-ABCs, neither Ashcroft nor hydroxyproline levels were increased in mice receiving A549 intratracheally. The figure shown below was included to the supplement as Fig. S5.

Figure 6. Airway basal cells of IPF patients (IPF-ABC) but not A549 augment fibrosis in bleomycin challenged NRG mice. NRG mice received either PBS (CTR) or bleomycin (Bleo) intratracheally and some mice (Bleo + IPF-ABC) additionally three days later 200,000 IPF-ABCs or A549 cells intratracheally. Lungs of mice were harvested for either histopathological scoring (Ashcroft, a, or hydroxyproline measurements b, CTR denotes for control mice; Bleo denotes for bleomycin; IPF-ABC denotes for airway basal cells derived from IPF patients. ANOVA with Tukey correction for multiple testing (a,b).

3-The control group should include a UIP-fibrotic lung disorder to compare with idiopathic-UIP/IPF, at least in some in vivo fundamental experiments. Indeed, the identification of a specific phenotype in IPF as compared with non UIP fibrotic ILD raise the important question of a UIP-specific or an IPF-specific ABC phenotype.

R4.3: We thank the reviewer for this comment and already performed some experiments in parallel with experiments testing the effect of IPF-ABCs. Indeed, we did not find any significant difference between IPF-ABCs or UIP-ABCs from patients with other ILD. Because we do not have further data on ABCs of non-IPF patients with a UIP-like pattern on HRCT, we prefer not to add the data to the manuscript, but added this important limitation to our discussion.

4- The identification of Src inhibition as a potential therapeutic target in lung fibrosis is not new (Hu, JPET 2014) and it is not surprising to see that Saracatinib inhibits lung fibrosis (Figure 6). The original result here is that Src activation might be specific to the ABC. However, the data provided by the authors to support that conclusion are limited. Indeed, immunofluorescence data in the fibrotic lung (Figure 4c and Figure S5) show only a few red dots on a faint red background (Fig S5). Interestingly, these data contrast with a strong and diffuse immunolabeling of the epithelial cells with immunohistochemistry. The discrepancy between methods is intriguing.

R4.4: We thank the reviewer for these comments and suggestions. The reviewer is right that src inhibition has been already tested in several models of organ fibrosis including the lungs. We didn't intend to neglect these previous literature and already cited the work of Hu et al. (J Pharmacol Exp Ther 2014, doi: 10.1124/jpet.114.216044) in our manuscript. In line with the reviewer's comment none of the published work on src inhibition in organ fibrosis focused on epithelial src expression. Our immunohistochemistry data support the concept that src is overexpressed not only by ABCs but also other cell types in IPF lungs. The reviewer is also right, that our originally submitted immunofluorescence staining was suboptimal and that the used polyclonal antibody produced a lot of background staining. To address the reviewer's comment we performed new IHC stainings using a different monoclonal antibody which produced much better results. The former immunofluorescence microphotographs were replaced (Fig. 4c).

5- Saracatinib is being used in the current manuscript as a src inhibitor. However, saracatinib is a SFK/Abl dual-kinase inhibitor. The Abl kinase is expressed in the lung epithelium and may be involved in lung repair after injury (Khatri, PNAS 2019, {Khatri, 2019 #1558}). Interpreting the pharmacological results should be cautious.

R4.5: We thank the reviewer for this comment and now state the fact of combined src and Abl inhibition by saracatinib clearly in our manuscript.

6-The statistical methods in the methods section is poor. The authors did use in some experiments a t-test where non parametric tests should be used due to the limited number of experiments performed.

R4.6: We thank the reviewer for this suggestion and performed a non-parametric test for the experiments outlined in Fig. 6 which analyze the effect of saracatinib treatment in our mouse model. The level of significance was not changed in these experiments by using the Mann Whitney test. The p-values and figure legends were changed accordingly. In accordance with the policy of the journal, for all statistical analyses we outline the name of the statistical test and type of comparison in the respective figure legend. We also revised the statistical methods section.

Minor comments

7-Figure 4b : please indicate the control and IPF tissue in the figure

R4.7: We thank the reviewer for this comment and added a text note to the figure.

8-Figure 4c and Figure S5: the authors state that c-src is highly expressed in the fibrotic lung. This is difficult to prove on the basis of the immunofluorescence pictures provided: this reviewer sees only a few red dots on a faint red background (Fig S5).

R4.8 As mentioned already in the reply to the reviewer`s fourth comment. The originally submitted immunofluorescence staining was suboptimal. To address the reviewer`s comment we performed new IHC stainings using a different monoclonal antibody which produced better results. The former immunofluorescence microphotographs were replaced.

9- Figure 6 : in the in vivo experiments, the effect of saracatinib on bleomycin induced lung fibrosis (without ABC injection) is missing.

R4.9 We thank the reviewer for this suggestion. In a parallel publication we studied the effect of saracatinib on bleomycin lung fibrosis. The respective manuscript is enclosed.

10-This reviewer would like to see the effect of src inhibition in human lung slices (control and IPF) as this is a unique full-human model, that would nicely complement the in vitro and in vivo experiments. This is important because the relevance of the organoid model to the pathophysiology of IPF is at best undetermined.

R4.10 We thank the reviewer for this great suggestion but think that adding such data is beyond the scope of the submitted manuscript. The effect of saracatinib using precision cut lung slices of IPF lungs will be addressed in an additional manuscript.

Reviewers' Comments:

Reviewer #1:

Remarks to the Author:

I thank the authors for their answers. They replied to almost all my questions, but I still have some concerns regarding the bronchosphere formation from HV-ABCs and IPF-ABCs (questions 5 and 8). Following on the authors answers, there are recent protocols that allow bronchosphere formation from healthy basal cells, even in the absence of feeders. Here is a review that describes different protocols that could be adapted to do so: Barkauskas CE et al. Development 2017. doi: 10.1242/dev.140103, so the authors are incorrect in saying that isolated HV-ABCs do not usually give rise to bronchospheres.

I still think it important to have a condition where the authors study bronchospheres from HV and IPF patients in parallel and compare their composition, differentiation and function. Even if the IPF bronchospheres grew better in their hands, it may be due to media composition favoring aberrant cell differentiation over healthy basal cells or that basal cells aggregated to form a sphere but didn't differentiate. Saying that this is a significant manuscript that covers a lot of ground with significant findings.

The experiments to address question 5 could be considered out of the scope of this article, so I suggest editing the discussion instead; lines 305-307: to add "in our conditions", line 328: to add "in the 3D culture model system we used", and a sentence discussing that HV-ABC can give rise to bronchospheres in other culture conditions.

However, I still believe data derived in answering question 8:- to understand differentiation (IHC/IF) of the IPF-ABC bronchospheres important; even if they can't be compared to HV-ABCs. Are these spheres simply a ball of basal cells? or have they managed to differentiate as one might expect from these organoids? Staining with differentiation markers would answer this.

Reviewer #2:

Remarks to the Author:

I am satisfy and donot have any comments on the revised manuscript.

Reviewer #3:

Remarks to the Author:

First of all, I appreciate the substantial efforts of the authors to revise the manuscript. Thanks to it, I think the manuscript has been very much improved. Especially, the biggest concern of mine on the rationale for effective use of the Src inhibitor, Saracatinib, which I have raised in the initial review, have been fully addressed by the attached paper. Honestly, I still have a remaining concern that thorough understanding of the relevance of this paper should be firstly fully rationalized by considering the two papers complementarily. However, I also consider that close mutual reference of the two papers should give a substantial impact for developing a new therapeutic strategy for this difficult disease. Indeed, I sincerely hope the authors continue further efforts to proceed with the clinical trial.

Reviewer #5:

Remarks to the Author:

The manuscript entitled "Airway Basal Cells show a dedifferentiated KRT17highPhenotype and promote Fibrosis in IPF" by Dr. Antje Prasse and colleagues reported characterization of airway basal cells in IPF. This is an important study to demonstrate a pathogenic role of IPF-ABCs in IPF pathogenesis. I agree with original reviewer 4's comments regarding some major methodological issues. Please note that I was not in previous round of review and was asked to primarily comment on the responses to reviewer 4's concerns.

R4.1. The response was okay. However, one should recognize that “brush cells” are apically extruded airway cells. It is surprised to me that the percentage of basal cells in the single cell RNA-seq data is so high - Figure 1a shows close to 50% airway basal cells. I was unable to find clear description if the cells from Figure 1a were directly from brushing or after culture.

R4.2. The response was suboptimal. The reviewer 4 asked for a control - any epithelial cells – AT2 cells or ciliated cells, or non-epithelial cells to demonstrate IPF-ABC-specific. A549 adenocarcinoma cells were instead used in the revision. I would suggest using flow sorted epithelial and non-epithelial fractions as better controls.

R4.3. The response was fine.

R4.4. The response was fine with a new figure 4C. Higher magnifications showing co-localization of SRC and ABC markers would help readers. Please note that SRC protein is ubiquitously expressed by most cell types.

R4.5. The response was fine. Please also note that ABL1 protein is ubiquitously expressed by most cell types.

R4.6. The response was fine.

Responses to minor comments were okay.

Additional comments:

1. Types and role of basal cells in IPF: It is interesting that the authors are trying to explain to reviewer 1 that IPF-ABCs are not basaloid cells, but they share some gene signatures. One would assume that IPF-ABCs contribute to the pool of pathogenetic “basal” cells in the interstitial areas of IPF lungs. To me, there is a lack of full demonstration of the differentiation potential of IPF-ABCs in comparison with HV-ABCs in this study. Perhaps, IPF-ABCs can differentiate to basaloid cells, especially under the influence of IPF fibroblasts. More staining for differentiation markers and transcriptomic analysis of spheres from IPF-ABCs and HV-ABC would help.

Point by Point Response to Reviewers' comments

REVIEWER COMMENTS

Re: NCOMMS-20-40313-2

Reviewer #1 (Remarks to the Author):

R1.1: I thank the authors for their answers. They replied to almost all my questions, but I still have some concerns regarding the bronchosphere formation from HV-ABCs and IPF-ABCs (questions 5 and 8). Following on the authors answers, there are recent protocols that allow bronchosphere formation from healthy basal cells, even in the absence of feeders. Here is a review that describes different protocols that could be adapted to do so: Barkauskas CE et al. Development 2017. doi: 10.1242/dev.140103, so the authors are incorrect in saying that isolated HV-ABCs do not usually give rise to bronchospheres.

I still think it important to have a condition where the authors study bronchospheres from HV and IPF patients in parallel and compare their composition, differentiation and function. Even if the IPF bronchospheres grew better in their hands, it may be due to media composition favoring aberrant cell differentiation over healthy basal cells or that basal cells aggregated to form a sphere but didn't differentiate. Saying that this is a significant manuscript that covers a lot of ground with significant findings. The experiments to address question 5 could be considered out of the scope of this article, so I suggest editing the discussion instead; lines 305-307: to add "in our conditions", line 328: to add "in the 3D culture model system we used", and a sentence discussing that HV-ABC can give rise to bronchospheres in other culture conditions.

AU1.1: We thank the reviewer for these thoughtful comments. We agree with the reviewer that in recent years the protocols for organoid/ bronchosphere generation of airway basal cells have been substantially refined by several groups which are highlighted well by Barkauskas CE et al. As mentioned by the reviewer, improved media and supplements have been developed, now allowing robust bronchosphere generation of HV-ABCs. We also think that our media composition may drive the described effects of IPF-ABCs. For our bronchosphere experiments, we used a basic, "old-fashioned" media composition that has been adopted from protocols used in oncology. We revised the respective sentences in our discussion as suggested by the reviewer.

In line with the reviewer, we are also convinced that the bronchosphere model is an optimal tool to study aberrant cell differentiation of IPF-ABCs.

R1.2: However, I still believe data derived in answering question 8:- to understand differentiation (IHC/IF) of the IPF-ABC bronchospheres important; even if they can't be compared to HV-ABCs. Are these spheres simply a ball of basal cells? or have they managed to differentiate as one might expect from these organoids? Staining with differentiation markers would answer this.

AU1.2: To address this question, we added further data obtained by confocal laser microscopy as a new Figure S3 to the supplement. Using "our conditions" we did not observe ABC differentiation

towards ciliated cells. After 21 days of culture, only few cells stained positive for the secretory cell marker MUC5AC. Almost all of the organoid cells stained positive for KRT5 and KRT17, which identifies them as basal cells. So, the reviewer is right that the IPF-ABCs aggregated to a sphere and did not further differentiate in our conditions. The immunofluorescence data were added to the supplement as Figure S3. In our 3D organoid culture up to day 21, we did not observe the emergence of aberrant basaloid cells. We agree with the reviewer that further studies are needed to study potential further dedifferentiation of IPF-ABCs towards aberrant basaloid cells in detail, including later time points.

Reviewer #2 (Remarks to the Author):

I am satisfy and do not have any comments on the revised manuscript.

Au 2.1: We thank the reviewer for the positive evaluation of our revised manuscript.

Reviewer #3 (Remarks to the Author):

First of all, I appreciate the substantial efforts of the authors to revise the manuscript. Thanks to it, I think the manuscript has been very much improved. Especially, the biggest concern of mine on the rationale for effective use of the Src inhibitor, Saracatinib, which I have raised in the initial review, have been fully addressed by the attached paper. Honestly, I still have a remaining concern that thorough understanding of the relevance of this paper should be firstly fully rationalized by considering the two papers complementarily. However, I also consider that close mutual reference of the two papers should give a substantial impact for developing a new therapeutic strategy for this difficult disease. Indeed, I sincerely hope the authors continue further efforts to proceed with the clinical trial.

Au 3.1: We thank the reviewer for the positive evaluation of our revised manuscript. We agree with the reviewer that the relevance of Src inhibition becomes more obvious if readers have access to both papers in parallel. The other manuscript was recently published on BioRxiv (doi: <https://doi.org/10.1101/2022.01.04.474955>). We added one sentence summarizing the main findings of this parallel paper to our discussion. The clinical trial is ongoing and more than half of the patient numbers are already recruited.

Reviewer #5 (Remarks to the Author):

The manuscript entitled "Airway Basal Cells show a dedifferentiated KRT17highPhenotype and promote Fibrosis in IPF" by Dr. Antje Prasse and colleagues reported characterization of airway basal cells in IPF. This is an important study to demonstrate a pathogenic role of IPF-ABCs in IPF pathogenesis. I agree with original reviewer 4's comments regarding some major methodological issues. Please note that I was not in previous round of review and was asked to primarily comment on the responses to reviewer 4's concerns.

R4.1. The response was okay. However, one should recognize that "brush cells" are apically extruded airway cells. It is surprised to me that the percentage of basal cells in the single cell RNA-seq data is

so high - Figure 1a shows close to 50% airway basal cells. I was unable to find clear description if the cells from Figure 1a were directly from brushing or after culture.

AU4.1: The cells used for the scRNAseq experiment were directly derived from brushings without further cell culture. The cell pellets obtained were frozen in DMSO/FCS and stored in nitrogen until the scRNA experiment. Freezing and thawing might have changed the original cell composition. It is likely that we thereby lost ciliated cells.

R4.2. The response was suboptimal. The reviewer 4 asked for a control - any epithelial cells – AT2 cells or ciliated cells, or non-epithelial cells to demonstrate IPF-ABC-specific. A549 adenocarcinoma cells were instead used in the revision. I would suggest using flow sorted epithelial and non-epithelial fractions as better controls.

AU4.2: As suggested by the reviewers, we performed the requested experiments using ciliated cells which were obtained from air liquid interface (ALI) cultures originally derived from IPF-ABCs. ALI cells were harvested after 1 month of culture and immediately sorted for PROM1+ cells by flow cytometry. Freshly sorted ciliated cells were then injected into NRG mice using the same protocol as previously used for ABCs or A549. Intratracheal injection of ciliated cells did not augment bleomycin-induced pulmonary fibrosis as detected by Ashcroft score or hydroxyproline levels. The data were added to Figure S6 in the supplement.

R4.3. The response was fine.

AU4.3: Thank you.

R4.4. The response was fine with a new figure 4C. Higher magnifications showing co-localization of SRC and ABC markers would help readers. Please note that SRC protein is ubiquitously expressed by most cell types.

AU4.4: We added additional images with higher magnifications as requested by the reviewer. The new images are added to figure S9.

R4.5. The response was fine. Please also note that ABL1 protein is ubiquitously expressed by most cell types.

AU4.5: We thank the reviewer for this helpful information.

R4.6. The response was fine. , Responses to minor comments were okay.

AU4.6: Thank you

Additional comments:

R5.1. Types and role of basal cells in IPF: It is interesting that the authors are trying to explain to reviewer 1 that IPF-ABCs are not basaloid cells, but they share some gene signatures. One would assume that IPF-ABCs contribute to the pool of pathogenetic “basal” cells in the interstitial areas of IPF lungs. To me, there is a lack of full demonstration of the differentiation potential of IPF-ABCs in comparison with HV-ABCs in this study. Perhaps, IPF-ABCs can differentiate to basaloid cells,

especially under the influence of IPF fibroblasts. More staining for differentiation markers and transcriptomic analysis of spheres from IPF-ABCs and HV-ABC would help.

AU5.1: We thank the reviewer for these additional comments. We also think that IPF-ABCs may give rise to aberrant basaloid cells. The transcriptional overlap is indeed huge. We are currently studying the potential further de-differentiation of IPF-ABCs towards aberrant basaloid cells in detail but, as mentioned in our rebuttal of reviewer 1, up to day 21 we did not observe any emergence of aberrant basaloid cells in our bronchosphere model based on IPF-ABCs derived from bronchial brushings.

Reviewers' Comments:

Reviewer #1:

Remarks to the Author:

Thank you for the further clarifications.

Reviewer #5:

Remarks to the Author:

My (and reviewer4's) concerns have been satisfactorily addressed. I have no further comments.

Point by Point Response to Reviewers' comments

REVIEWER COMMENTS

Re: NCOMMS-20-40313B

Reviewer #1 (Remarks to the Author):

R1.1: Thank you for the further clarifications.

Au 1.1: We thank the reviewer for the positive evaluation of our revised manuscript.

Reviewer #5 (Remarks to the Author):

R5.1: My (and reviewer4's) concerns have been satisfactorily addressed. I have no further comments.

Au 5.1: We thank the reviewer for the positive evaluation of our revised manuscript.